# ADAPTIVE BUDGET ALLOCATION FOR PARAMETER-EFFICIENT FINE-TUNING

**Qingru Zhang**[†][*]**, Minshuo Chen**[‡]**, Alexander Bukharin**[†]**, Pengcheng He**[◇]**, Yu Cheng**[◇]**, Weizhu Chen**[◇] **and Tuo Zhao**[†]
[†]Georgia Institute of Technology  [‡]Princeton University  [◇]Microsoft Azure AI
{qingru.zhang,abukharin3,tourzhao}@gatech.edu
mc0750@princeton.edu
{penhe,yu.cheng,wzchen}@microsoft.com

## ABSTRACT

Fine-tuning large pre-trained language models on downstream tasks has become an important paradigm in NLP. However, common practice fine-tunes all of the parameters in a pre-trained model, which becomes prohibitive when a large number of downstream tasks are present. Therefore, many fine-tuning methods are proposed to learn incremental updates of pre-trained weights in a parameter efficient way, e.g., low-rank increments. These methods often evenly distribute the budget of incremental updates across all pre-trained weight matrices, and overlook the varying importance of different weight parameters. As a consequence, the fine-tuning performance is suboptimal. To bridge this gap, we propose AdaLoRA, which adaptively allocates the parameter budget among weight matrices according to their importance score. In particular, AdaLoRA parameterizes the incremental updates in the form of singular value decomposition. Such a novel approach allows us to effectively prune the singular values of unimportant updates, which is essentially to reduce their parameter budget but circumvent intensive exact SVD computations. We conduct extensive experiments with several pre-trained models on natural language processing, question answering, and natural language generation to validate the effectiveness of AdaLoRA. Results demonstrate that AdaLoRA manifests notable improvement over baselines, especially in the low budget settings. Our code is publicly available at https://github.com/QingruZhang/AdaLoRA.

## 1 INTRODUCTION

Pre-trained language models (PLMs) have manifested superior performance in various natural language processing tasks (Devlin et al., 2019; Liu et al., 2019; He et al., 2021b; Radford et al., 2019; Brown et al., 2020). The most common way to adapt pre-trained models to down-stream tasks is to fine-tune all the parameters (full fine-tuning, Qiu et al. (2020); Raffel et al. (2020)). However, pre-trained models typically incurs large memory footprint. For example, BERT model (Devlin et al., 2019) consists up to 300 million parameters; T5 (Raffel et al., 2020) comprises up to 11 billion parameters and GPT-3 (Brown et al., 2020) contains up to 175 billion parameters. When building a NLP system upon these pre-trained models, we usually handle multiple tasks that arrive simultaneously (Radford et al., 2019). Given a large number of down-stream tasks, full fine-tuning requires that each task maintains a separated copy of large models. The resulting memory consumption is prohibitively expensive.

To address this issue, researchers have proposed two main lines of research to reduce the fine-tuning parameters, while maintaining or even improving the performance of PLMs. Specifically, one line of research focuses on adding small neural modules to PLMs and fine-tune only these modules for each task – the base model is kept frozen and shared across tasks. In this way, only a small number of task-specific parameters are introduced and updated, greatly enhancing the practicality of large models. For example, adapter tuning (Houlsby et al., 2019; Rebuffi et al., 2017; Pfeiffer et al., 2020;

---

[*]Work was done during Qingru Zhang's internship at Microsoft Azure AI.

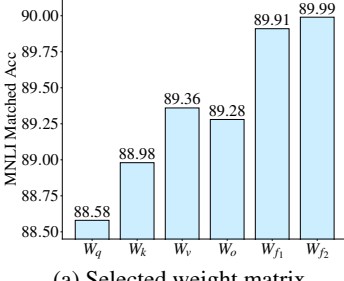
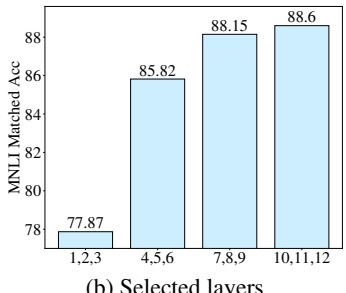

| (a) Selected weight matrix | (b) Selected layers |

Figure 1: Given the total trainable parameters as 0.28M, we apply LoRA only to selected weight matrices (left) or selected layers (right) of DeBERTaV3-base and compare the fine-tuning performance on MNLI-m. Figure 1a: we only fine-tune a selected type of weight matrix of every transformer layer, including query/key/value projection ($W_q, W_k, W_v$), output projection ($W_o$) in the self-attention, and two weight matrices ($W_{f_1}, W_{f_2}$) in two-layer FFNs. In Figure 1b, we apply LoRA to every weight matrix of the selected layers.

He et al., 2022) inserts small neural modules called adapters between the layers of the base model. Prefix tuning (Li & Liang, 2021) and prompt tuning (Lester et al., 2021) attach additional trainable prefix tokens to the input or hidden layers of the base model. These methods have shown to achieve comparable performance to full fine-tuning, while only updating less than $1\%$ of the original model parameters, significantly releasing the memory consumption.

Another line of research proposes to model the incremental update of the pre-trained weights in a parameter-efficient way, without modifying the model architecture (Zaken et al., 2021; Guo et al., 2020; Hu et al., 2022). Given a pre-trained weight matrix[1] $W^{(0)}$, for example, diff pruning (Guo et al., 2020) models its incremental update $\Delta$ as a sparse matrix. Diff pruning initializes $\Delta$ as the same dimension as $W^{(0)}$ and then prunes $\Delta$ element-wise based on the magnitude of the entries. As such, diff pruning can increase the parameter efficiency substantially by adaptively retaining important updates and pruning unimportant ones. Nonetheless, diff pruning has several limitations. First, it relies on low-level implementation to speed up the computation of unstructured sparse matrices, which is not well supported by existing deep learning frameworks. Therefore, we have to store $\Delta$ as a dense matrix during training. Second, it needs to update every entry of $\Delta$ with their gradients and then prune them. This results in similar computational cost as full fine-tuning (Guo et al., 2020).

To overcome these drawbacks, Hu et al. (2022) propose a method named LoRA, which parameterizes $\Delta$ as a low-rank matrix by the product of two much smaller matrices:

$$W = W^{(0)} + \Delta = W^{(0)} + BA, \qquad (1)$$

where $W^{(0)}, \Delta \in \mathbb{R}^{d_1 \times d_2}$, $A \in \mathbb{R}^{r \times d_2}$ and $B \in \mathbb{R}^{d_1 \times r}$ with $r \ll \{d_1, d_2\}$. During fine-tuning, only $A$ and $B$ are updated. The rank $r$ is chosen to be much smaller than the dimension of $W$ (e.g., $r = 8$ when $d_1 = d_2 = 1024$). With less than $0.5\%$ additional trainable parameters, the training overhead can be reduced up to $70\%$, compared to full fine-tuning. However, LoRA achieves comparable or even better performance than full fine-tuning (Hu et al., 2022). Meanwhile, the product of two samll matrices is more friendly to implement and deploy than unstructured sparse matrices in diff pruning.

LoRA still has limitations as it prespecifies the rank $r$ of each incremental matrix $\Delta$ identical. This ignores the fact that the importance of weight matrices varies significantly across modules and layers when fine-tuning pre-trained models. To illustrate this point, we present an concrete example in Figure 1. We compare the performance of LoRA when fine-tuning specific modules or layers with the same number of trainable parameters. Figure 1a shows that fine-tuning feed-forward networks (FFN) achieves better performance than self-attention modules. In addition, Figure 1b demonstrates that weight matrices in top layers are more important than those in bottom layers.

Adding more trainable parameters to the critical weight matrices can lead to better model performance. In contrast, adding more parameters to those less important weight matrices yields very marginal gains or even hurt model performance. Given the parameter budget, i.e., the number of total trainable parameters, we always prefer to allocate more parameters to those important modules. Distributing the budget evenly to all weight matrices/layers, like LoRA and other methods (e.g., adapter and prefix tuning), often gives suboptimal performance. To this end, a natural question is:

> ***How can we allocate the parameter budget adaptively according to importance of modules to improve the performance of parameter-efficient fine-tuning?***

---

[1]Unless specified otherwise, we use $W^{(0)}$ to denote any pre-trained weight matrix.

To answer this question, we propose a new method – *AdaLoRA* (Adaptive Low-Rank Adaptation), which dynamically allocates the parameter budget among weight matrices during LoRA-alike fine-tuning. Specifically, AdaLoRA adjusts the rank of incremental matrices to control their budget. Critical incremental matrices are assigned with high rank such that they can capture more fine-grained and task-specific information. Less importance ones are pruned to have lower rank to prevent overfitting and save the computational budget. There are some methods to control the rank of matrices in the existing literature of matrix approximation (Cai et al., 2010; Koltchinskii et al., 2011; Toh & Yun, 2010). Most of them directly compute singular value decomposition (SVD) of a matrix and then truncate the smallest singular values. Such an operation can manipulate the rank explicitly and, more importantly, minimize the difference between the resulting matrix and the original matrix. However, for fine-tuning large models, it becomes prohibitively expensive to iteratively apply SVD for a large number of high-dimensional weight matrices. Therefore, instead of computing SVD exactly, we parameterize $\Delta$ as $\Delta = P\Lambda Q$ to mimic SVD. The diagonal matrix $\Lambda$ contains singular values while the orthogonal matrices $P$ and $Q$ represent left/right singular vectors of $\Delta$. To regularize the orthogonality of $P$ and $Q$, an additional penalty is added to training loss. Such a parameterization avoids the intensive computations of SVD. Besides, another advantage is that we only need to drop the unimportant singular values while the singular vectors are maintained. This preserves the possibility of future recovery and stabilizes the training. See a detailed comparison to LoRA in Section 3.

Based on our SVD parameterization, AdaLoRA dynamically adjusts the rank of $\Delta = PVQ$ by *importance scoring*. Specifically, we divide the incremental matrix $P\Lambda Q$ into triplets, where each triplet $\mathcal{G}_i$ contains the $i$-th singular value and the corresponding singular vectors. To quantify the importance of triplets, we propose a novel importance metric, which takes account of the contribution of every entry in $\mathcal{G}_i$ to the model performance (Sanh et al., 2020; Liang et al., 2021; Zhang et al., 2022). Triplets with low importance scores are granted low priority and hence the singular values are zeroed out. Triplets with high importance are retained for fine-tuning. Moreover, we also propose a global budget scheduler to facilitate the training. In particular, we start from an initial parameter budget, which is slightly higher than the final budget, and then gradually reduce it until matching the target. Such a scheduler can improve the training stability and model performance. Please see Section 3 for a detailed description of our importance metric and budget scheduler.

We conduct extensive experiments on a wide range of tasks and models to demonstrate the effectiveness of AdaLoRA. Specifically, we evaluate the performance using DeBERTaV3-base (He et al., 2021a) on natural language understanding (GLUE, Wang et al. (2019)) and question answering (SQuADv1, Rajpurkar et al. (2016) and SQuADv2, Rajpurkar et al. (2018)) datasets. We also apply our methods to BART-large (Lewis et al., 2019) and evaluate the performance on natural language generation (XSum, Narayan et al. (2018) and CNN/DailyMail, Hermann et al. (2015)) tasks. We show AdaLoRA consistently outperforms the baseline, especially under low budget settings. For example, with less than $0.1\%$ trainable parameters of full fine-tuning, AdaLoRA achieves a 1.2% F1 improvement on the SQuAD2.0 dataset compared with state-of-the-art approaches.

## 2 BACKGROUND

**Transformer-based Models.** A typical transformer model consists of $L$ stacked blocks, where each block contains two submodules: a multi-head attention (MHA) and a fully connected FFN. Given the input sequence $X \in \mathbb{R}^{n \times d}$, MHA performs the attention function in parallel $h$ heads:

$$\text{MHA}\left(X\right) = \text{Concat}(\text{head}_1, ..., \text{head}_h)W_o, \quad \text{head}_i = \text{Softmax}\left(XW_{q_i}(XW_{k_i})^\top / \sqrt{d_h}\right)XW_{v_i},$$

where $W_o \in \mathbb{R}^{d \times d}$ is an output projection and $W_{q_i}, W_{k_i}, W_{v_i} \in \mathbb{R}^{d \times d_h}$ are query, key and value projections of head $i$. $d_h$ is typically set to $d/h$. The other important module is a FFN which consists of two linear transformations with a ReLU activation in between: $\text{FFN}(X) = \text{ReLU}(XW_{f_1} + \boldsymbol{b}_1)W_{f_2} + \boldsymbol{b}_2$, where $W_{f_1} \in \mathbb{R}^{d \times d_m}$ and $W_{f_2} \in \mathbb{R}^{d_m \times d}$. Finally, a residual connection is used followed by a layer normalization (Ba et al., 2016).

**Low Rank Adaptation.** LoRA (Hu et al., 2022) models the incremental update of the pre-trained weights by the product of two small matrices. For $\boldsymbol{h} = W^{(0)}\boldsymbol{x}$, the modified forward pass is:

$$\boldsymbol{h} = W^{(0)}\boldsymbol{x} + \Delta\boldsymbol{x} = W^{(0)}\boldsymbol{x} + BA\boldsymbol{x}, \tag{2}$$

where $W^{(0)}, \Delta \in \mathbb{R}^{d_1 \times d_2}$, $A \in \mathbb{R}^{r \times d_2}$ and $B \in \mathbb{R}^{d_1 \times r}$ with $r \ll \{d_1, d_2\}$. $A$ typically adopts a random Gaussion initialization while $B$ is initialized with zero to have $\Delta = 0$ at the beginning of

training. We further denote $A_{i*}$ as the $i$-th row of $A$, $B_{*i}$ as the $i$-th column of $B$, and $\mathcal{G}_i = \{A_{i*}, B_{*i}\}$ as the $i$-th doublet. Hu et al. (2022) only apply LoRA to query and value projections (i.e, $W_q$ and $W_v$) in the MHAs. He et al. (2022) extend it to weight matrices of FFNs (i.e, $W_{f_1}$ and $W_{f_2}$), leading to the performance improvement . Meanwhile, they propose a unified view of various efficient tuning methods including adapter tuning, prefix tuning and LoRA.

# 3 ADALoRA METHOD

Our method contains two important components: (i) SVD-based adaptation, which formulates the incremental matrices in the form of singular value decomposition; (ii) Importance-aware rank allocation, which prunes redundant singular values based on our newly-designed importance metric.

## 3.1 SVD-BASED ADAPTATION

As mentioned in Section 1, we propose to parameterize the incremental updates of the pre-trained weight matrices in the form of singular value decomposition:

$$W = W^{(0)} + \Delta = W^{(0)} + P\Lambda Q, \tag{3}$$

where $P \in \mathbb{R}^{d_1 \times r}$ and $Q \in \mathbb{R}^{r \times d_2}$ represent the left/right singular vectors of $\Delta$ and the diagonal matrix $\Lambda \in \mathbb{R}^{r \times r}$ contains the singular values $\{\lambda_i\}_{1 \le i \le r}$ with $r \ll \min(d_1, d_2)$. We further denote $\mathcal{G}_i = \{P_{*i}, \lambda_i, Q_{i*}\}$ as the triplet containing the $i$-th singular value and vectors. In practice, since $\Lambda$ is diagonal, we only need to save it as a vector in $\mathbb{R}^r$. $\Lambda$ is initialized with zero while $P$ and $Q$ adopt a random Gaussian initialization to ensure $\Delta = 0$ at the beginning of training. To enforce the orthogonality of $P$ and $Q$, i.e., $P^\top P = QQ^\top = I$, we utilize the following regularizer[2]:

$$R(P, Q) = \|P^\top P - I\|_F^2 + \|QQ^\top - I\|_F^2. \tag{4}$$

In our method, $\Lambda$ is iteratively pruned to adjust the rank after each gradient decent step. As mentioned in Section 1, one can directly compute SVD for every $\Delta$ to manipulate singular values. The computational complexity, however, is $O(\min(d_1, d_2)d_1 d_2)$. It becomes extremely expensive to iteratively apply SVD for a large number of high-dimensional incremental matrices. In contrast, our parameterization avoids intensive SVD computation, greatly releasing the computational overhead.

We remark that one can also apply structured pruning to LoRA to control the rank (i.e., prune $BA$ doublet-wise in (1)), whereas it has the following disadvantages. First, when a doublet is measured as unimportant, we have to prune all of its elements. It makes scarcely possible to reactivate the pruned doublets as their entries are all zeroed out and not trained. In contrast, AdaLoRA only masks out the singular values based on (3) while the singular vectors are always maintained. It preserves the potential of future recovery for the triplets dropped by mistake. Second, $A$ and $B$ of LoRA are not orthogonal, meaning the doublets can be dependent with each other. Discarding the doublets can incur larger variation from the original matrix than truncating the smallest singular values. Therefore, the incremental matrices are often altered dramatically after each step of rank allocation, which causes training instability and even hurts generalization. To demonstrate this point, we present an ablation study in Section 4.4, which compares AdaLoRA with structured pruning for LoRA.

## 3.2 IMPORTANCE-AWARE RANK ALLOCATION

We apply the SVD-based adaptation (3) to every weight matrix including $W_q$, $W_k$, $W_v$, $W_{f_1}$ and $W_{f_2}$ of each transformer layer. In order to control the budget, we iteratively prune singular values in correspondence to their importance score during the training. For clear reference, we use $k$ to index the incremental matrix, i.e., $\Delta_k = P_k \Lambda_k Q_k$ for $k = 1, \ldots, n$, where $n$ is the number of adapted weight matrices. We denote the $i$-th triplet of $\Delta_k$ as $\mathcal{G}_{k,i} = \{P_{k,*i}, \lambda_{k,i}, Q_{k,i*}\}$ and its importance score as $S_{k,i}$. We further denote the parameter sets $\mathcal{P} = \{P_k\}_{k=1}^n$, $\mathcal{E} = \{\Lambda_k\}_{k=1}^n$, $\mathcal{Q} = \{Q_k\}_{k=1}^n$ and training cost as $\mathcal{C}(\mathcal{P}, \mathcal{E}, \mathcal{Q})$. With the regularization (4), the training objective is given by $\mathcal{L}(\mathcal{P}, \mathcal{E}, \mathcal{Q}) = \mathcal{C}(\mathcal{P}, \mathcal{E}, \mathcal{Q}) + \gamma \sum_{k=1}^n R(P_k, Q_k)$, where $\gamma > 0$ is the regularization coefficient. At the $t$-th step, we first take a stochastic gradient step to update $P_k^{(t)}, \Lambda_k^{(t)}$ and $Q_k^{(t)}$ for $k = 1, \ldots, n$. Specifically, for $\Lambda_k^{(t)}$

$$\tilde{\Lambda}_k^{(t)} = \Lambda_k^{(t)} - \eta \nabla_{\Lambda_k} \mathcal{L}(\mathcal{P}^{(t)}, \mathcal{E}^{(t)}, \mathcal{Q}^{(t)}), \tag{5}$$

---

[2]We present the experiments in Appendix I to verify the effectiveness of the regularization.

where $\eta > 0$ is learning rate. Then, given importance score $S_k^{(t)}$, the singular values are pruned following

$$\Lambda_k^{(t+1)} = \mathcal{T}(\tilde{\Lambda}_k^{(t)}, S_k^{(t)}), \text{ with } \mathcal{T}(\tilde{\Lambda}_k^{(t)}, S_k^{(t)})_{ii} = \begin{cases} \tilde{\Lambda}_{k,ii}^{(t)} & S_{k,i}^{(t)} \text{ is in the top-}b^{(t)} \text{ of } S^{(t)}, \\ 0 & \text{otherwise}, \end{cases} \quad (6)$$

where $S^{(t)} = \{S_{k,i}^{(t)}\}_{1 \le k \le n, 1 \le i \le r}$ contains the importance score of all triplets. Here $b^{(t)}$ is the budget of remaining singular values at the $t$-th step, which we explain more in Section 3.3. In this way, we leave more budget to the incremental matrices of higher priority by pruning the singular values of less important ones. In the sequel, we introduce several options to design the importance score.

**Magnitude of singular values** is the most straightforward way to quantify the importance of every triplet, i.e., $S_{k,i} = |\lambda_{k,i}|$. In this way, only the least significant singular values are discarded. It minimizes the deviation from the original matrix and further stabilizes the training. Many existing methods use this criterion to control the rank of matrix (Cai et al., 2010; Koltchinskii et al., 2011; Toh & Yun, 2010). However, we remark that such a simple metric cannot properly quantify the contribution of parameters to model performance.

**Sensitivity-based importance** is another option for importance scoring, which quantifies the sensitivity of parameters to the training loss (Molchanov et al., 2019; Sanh et al., 2020; Liang et al., 2021; Zhang et al., 2022). The prior work, however, leverages the sensitivity to quantify the importance of single entries and applies it for unstructured pruning that prunes weights element-wise. When it turns to our case, we have to design a new metric as the triplets are discarded group-wise. Every entry's sensitivity ought to be considered and properly combined to quantify the overall contribution of the triplet to model performance. Therefore, we propose a newly-designed importance metric in account of both the singular value and vectors in triplet $\mathcal{G}_{k,i}$:

$$S_{k,i} = s(\lambda_{k,i}) + \frac{1}{d_1} \sum_{j=1}^{d_1} s(P_{k,ji}) + \frac{1}{d_2} \sum_{j=1}^{d_2} s(Q_{k,ij}), \quad (7)$$

where we calculate the mean importance of $P_{k,*i}$ and $Q_{k,i*}$ such that $S_{k,i}$ does not scale with the number of parameters in $\mathcal{G}_{k,i}$. Here $s(\cdot)$ is a specific importance function for single entries. We can adopt the sensitivity for $s(\cdot)$, which is defined as the magnitude of the gradient-weight product:

$$I(w_{ij}) = |w_{ij} \nabla_{w_{ij}} \mathcal{L}|, \quad (8)$$

where $w_{ij}$ is any trainable parameter. (8) essentially approximates the change in loss when a parameter is zeroed out. If the removal of a parameter has a large influence, then the model is sensitive to it and we should retain it (Molchanov et al., 2019; Liang et al., 2021; Zhang et al., 2022).

However, Zhang et al. (2022) point out that the sensitivity in (8) is not yet a reliable importance indicator. Such a score is estimated on the sampled mini batch. The stochastic sampling and complicated training dynamics incur high variability and large uncertainty for estimating the sensitivity with (8). Therefore, Zhang et al. (2022) propose to resolve this issue by sensitivity smoothing and uncertainty quantification:

$$\overline{I}^{(t)}(w_{ij}) = \beta_1 \overline{I}^{(t-1)}(w_{ij}) + (1 - \beta_1) I^{(t)}(w_{ij}) \quad (9)$$

$$\overline{U}^{(t)}(w_{ij}) = \beta_2 \overline{U}^{(t-1)}(w_{ij}) + (1 - \beta_2) \left| I^{(t)}(w_{ij}) - \overline{I}^{(t)}(w_{ij}) \right|, \quad (10)$$

where $0 < \beta_1, \beta_2 < 1$. $\overline{I}^{(t)}$ is the smoothed sensitivity by exponential moving average and $\overline{U}^{(t)}$ is the uncertainty term quantified by the local variation between $I^{(t)}$ and $\overline{I}^{(t)}$. Then they define the importance as the product between $\overline{I}^{(t)}$ and $\overline{U}^{(t)}$, which can be another option for $s(\cdot)$:

$$s^{(t)}(w_{ij}) = \overline{I}^{(t)}(w_{ij}) \cdot \overline{U}^{(t)}(w_{ij}). \quad (11)$$

We present a detailed ablation study in Section 4.4 to compare the performance of different importance metrics. We find the proposed metric (7) based on the sensitivity variant (11) generally performs best. We summarize the detailed algorithm in Algorithm 1 presented in Appendix A.

### 3.3 GLOBAL BUDGET SCHEDULER

As mentioned in Section 1, adjusting the rank is naturally to control the parameter budget in the context of low-rank adaptation. Hence we define the budget $b^{(t)}$ as the total rank of all incremental matrices, i.e., the number of total singular values. Recall that the budget allocation is iteratively conducted during the fine-tuning. To facilitate the training, we propose a global budget scheduler. Specifically, we start from an initial budget $b^{(0)}$ that is slightly higher than the target budget $b^{(T)}$ (e.g., 1.5 times of $b^{(T)}$). We set the initial rank of each incremental matrix as $r = b^{(0)}/n$. We warm up the training for $t_i$ steps, and then follow a cubic schedule to decrease the budget $b^{(t)}$ until it reaches $b^{(T)}$. Finally, we fix the resulting budget distribution and fine-tune the model for $t_f$ steps. The exact equation for the budget schedule is presented in Appendix B. This allows AdaLoRA to explore the parameter space first and then focus on the most important weights later.

## 4 EXPERIMENTS

We implement AdaLoRA for fine-tuning DeBERTaV3-base (He et al., 2021a) and BART-large (Lewis et al., 2019). We evaluate the effectiveness of the proposed algorithm on natural language understanding (GLUE, Wang et al. (2019)), question answering (SQuADv1, Rajpurkar et al. (2016) and SQuADv2, Rajpurkar et al. (2018)), and natural language generation (XSum, Narayan et al. (2018) and CNN/DailyMail Hermann et al. (2015)). All the gains have passed significant tests with $p < 0.05$.

**Implementation Details.** We use *PyTorch* (Paszke et al., 2019) to implement all the algorithms. Our implementation is based on the publicly available *Huggingface Transformers*[3] (Wolf et al., 2019) code-base. All the experiments are conducted on NVIDIA V100 GPUs.

LoRA scales $\Delta x$ by $\alpha/r$ where $\alpha$ is a constant in $r$. As a result, the magnitude of output can be consistent given different $r$. It reduces the efforts of retuning learning rate when varying $r$. Typically $\alpha$ is set as 16 or 32 and never tuned (Hu et al., 2022; Yang & Hu, 2020). Following LoRA, we add the same scaling for (3) and fix $\alpha$ as LoRA. Besides, in Algorithm 1, we prune singular values every $\Delta_T$ steps (e.g., $\Delta_T = 100$) such that the pruned triplets can still get updated within these intervals and possibly reactivated in future iterations.

**Baselines.** We compare AdaLoRA with the following methods:

• *Full fine-tuning* is the most common approach for adaptation. During fine-tuning, the model is initialized with pre-trained weights and biases, and all model parameters undergo gradient updates.

• *Bitfit* (Zaken et al., 2021) is an effective parameter-efficient fine-tuning method. The method only fine-tunes bias vectors in the pre-trained model.

• *Adapter tuning* (Houlsby et al., 2019; Pfeiffer et al., 2020) inserts two-layer adapters between transformer blocks. We compare with two types of adapter. *Houlsby adapter* as proposed in Houlsby et al. (2019) is inserted between the self-attention module and the FFN module followed by a subsequent residual connection. Recently, Pfeiffer et al. (2020) propose a more efficient design with adapters only applied after FFN modules and LayerNorm modules (Ba et al., 2016), which we call *Pfeiffer adapter*. The number of trainable parameters is determined by the number of layers, the hidden dimension of adapters and the dimension of their inputs.

• *LoRA* (Hu et al., 2022) is a state-of-the-art method for parameter-efficient fine-tuning. The method parameterizes incremental updates by two small matrices and only fine-tune them. The number of trainable parameter is controlled by the rank $r$ and the number of adapted weight matrices $n$. Hu et al. (2022) apply LoRA to query and value projections only. In empirical, we find that applying LoRA to all weight matrices, i.e., $W_q, W_k, W_v, W_{f_1}$ and $W_{f_2}$, can further improve its performance (Please see Appendix H). Hence, we compare with this generalized LoRA to maximize its performance. We use publicly available implementation [4] to run all the baselines. Please refer to Hu et al. (2022) and reference therein for details.

---

[3] https://github.com/huggingface/transformers
[4] https://github.com/microsoft/LoRA

Table 1: Results with DeBERTaV3-base on GLUE development set. The best results on each dataset are shown in **bold**. We report the average correlation for STS-B. *Full FT*, *HAdapter* and *PAdapter* represent full fine-tuning, Houlsby adapter, and Pfeiffer adapter respectively. We report mean of 5 runs using different random seeds.

| Method | # Params | MNLI m/mm | SST-2 Acc | CoLA Mcc | QQP Acc/F1 | QNLI Acc | RTE Acc | MRPC Acc | STS-B Corr | All Ave. |
|---|---|---|---|---|---|---|---|---|---|---|
| Full FT | 184M | 89.90/90.12 | 95.63 | 69.19 | **92.40/89.80** | 94.03 | 83.75 | 89.46 | 91.60 | 88.09 |
| BitFit | 0.1M | 89.37/89.91 | 94.84 | 66.96 | 88.41/84.95 | 92.24 | 78.70 | 87.75 | 91.35 | 86.02 |
| HAdapter | 1.22M | 90.13/90.17 | 95.53 | 68.64 | 91.91/89.27 | 94.11 | 84.48 | 89.95 | 91.48 | 88.12 |
| PAdapter | 1.18M | 90.33/90.39 | 95.61 | 68.77 | 92.04/89.40 | 94.29 | 85.20 | 89.46 | 91.54 | 88.24 |
| LoRA$_{r=8}$ | 1.33M | 90.65/90.69 | 94.95 | 69.82 | 91.99/89.38 | 93.87 | 85.20 | 89.95 | 91.60 | 88.34 |
| AdaLoRA | 1.27M | **90.76/90.79** | **96.10** | **71.45** | 92.23/89.74 | **94.55** | **88.09** | **90.69** | **91.84** | **89.31** |
| HAdapter | 0.61M | 90.12/90.23 | 95.30 | 67.87 | 91.65/88.95 | 93.76 | 85.56 | 89.22 | 91.30 | 87.93 |
| PAdapter | 0.60M | 90.15/90.28 | 95.53 | 69.48 | 91.62/88.86 | 93.98 | 84.12 | 89.22 | 91.52 | 88.04 |
| HAdapter | 0.31M | 90.10/90.02 | 95.41 | 67.65 | 91.54/88.81 | 93.52 | 83.39 | 89.25 | 91.31 | 87.60 |
| PAdapter | 0.30M | 89.89/90.06 | 94.72 | 69.06 | 91.40/88.62 | 93.87 | 84.48 | 89.71 | 91.38 | 87.90 |
| LoRA$_{r=2}$ | 0.33M | 90.30/90.38 | 94.95 | 68.71 | 91.61/88.91 | 94.03 | 85.56 | 89.71 | **91.68** | 88.15 |
| AdaLoRA | 0.32M | **90.66/90.70** | **95.80** | **70.04** | **91.78/89.16** | **94.49** | **87.36** | **90.44** | 91.63 | **88.86** |

## 4.1 NATURAL LANGUAGE UNDERSTANDING

**Models and Datasets.** We evaluate the fine-tuning performance of DeBERTaV3-base (He et al., 2021a) using the proposed algorithm. We conduct experiments on the General Language Understanding Evaluation (GLUE, Wang et al. 2019) benchmark. The benchmark includes two single-sentence classification tasks, three similarity and paraphrase tasks and four natural language inference tasks. Dataset details are summarized in Appendix D.

**Implementation Details.** DeBERTaV3-base consists of 183 millions parameters. We compare AdaLoRA with the baselines under different budget levels, for example, given the total trainable parameters as 0.3/0.6/1.2 million. In order to match the parameter budget, we select the hidden dimensions of adapters from $\{8, 16, 32, 64\}$, set the rank $r$ of LoRA as $\{2, 4, 8\}$, and choose the final budget $b^{(T)}$ of AdaLoRA from $\{144, 288, 576\}$. Then we set $b^{(0)}$ as 1.5 times of $b^{(T)}$ for AdaLoRA and select the regularization coefficient $\gamma$ from $\{0.1, 0.3, 0.5\}$. We set the exponential moving average parameters $\beta_1$ and $\beta_2$ as their default value 0.85. We select the learning rate from $\{5 \times 10^{-5}, 8 \times 10^{-5}, 1 \times 10^{-4}, 2 \times 10^{-4}\}$. More details are presented in Appendix E.

**Main results.** We compare AdaLoRA with the baseline methods under different budget settings. Table 1 shows experimental results on the GLUE development set. We see that AdaLoRA achieves better or on par performance compared with existing approaches on all datasets under all budget levels. For example, when the parameter budget is 0.3M, AdaLoRA achieves 87.36% accuracy on RTE, which is 1.8% higher than the best-performing baseline. Besides, AdaLoRA with extreme low budget can often perform better than the baselines with higher budget. For example, AdaLoRA achieve 70.04% Mcc. score on CoLA with 0.3M fine-tuning parameters, which is higher than all baseline methods with lager budget (e.g., 0.6M and 1.2M).

## 4.2 QUESTION ANSWERING

**Models and Datasets.** We evaluate performance of the proposed algorithm on two question answering (QA) datasets: SQuAD v1.1 (Rajpurkar et al., 2016) and SQuADv2.0 (Rajpurkar et al., 2018), where we use AdaLoRA to fine-tune DeBERTaV3-base. These tasks are treated as a sequence labeling problem, where we predict the probability of each token being the start and end of the answer span. Dataset details can be found in Appendix F.

**Implementation Details.** We compare AdaLoRA with the baseline methods under different parameter budgets. That is we have the number of trainable parameters as $0.08\%/0.16\%/0.32\%/0.65\%$ of total pre-trained parameters. To match the budget requirements, we select the hidden dimensions of adapters from $\{4, 8, 16, 32, 64\}$, set the rank $r$ of LoRA as $\{1, 2, 4, 8\}$ and choose the final total rank $b^{(T)}$ of AdaLoRA from $\{72, 144, 288, 576\}$. We set the batch size as 16. We use AdamW (Loshchilov & Hutter, 2019) as the optimizer and we set the learning rate as $1 \times 10^{-3}$ for AdaLoRA. Please refer to Appendix F for more details.

**Main Results.** Table 2 summarizes experimental results when we fine-tune DeBERTaV3-base under 4 different budget settings: 0.08%, 0.16%, 0.32% and 0.65% of total pre-trained parameters. From the

Table 2: Results with DeBERTaV3-base on SQuAD v1.1 and SQuADv2.0. Here *# Params* is the number of trainable parameters relative to that in full fine-tuning. We report EM/F1. The best results in each setting are shown in **bold**.

| | SQuADv1.1 | | | | SQuADv2.0 | | | |
|---|---|---|---|---|---|---|---|---|
| Full FT | 86.0 / 92.7 | | | | 85.4 / 88.4 | | | |
| # Params | 0.08% | 0.16% | 0.32% | 0.65% | 0.08% | 0.16% | 0.32% | 0.65% |
| HAdapter | 84.4/91.5 | 85.3/92.1 | 86.1/92.7 | 86.7/92.9 | 83.4/86.6 | 84.3/87.3 | 84.9/87.9 | 85.4/88.3 |
| PAdapter | 84.4/91.7 | 85.9/92.5 | 86.2/92.8 | 86.6/93.0 | 84.2/87.2 | 84.5/87.6 | 84.9/87.8 | 84.5/87.5 |
| LoRA | 86.4/92.8 | 86.6/92.9 | 86.7/93.1 | 86.7/93.1 | 84.7/87.5 | 83.6/86.7 | 84.5/87.4 | 85.0/88.0 |
| AdaLoRA | **87.2/93.4** | **87.5/93.6** | **87.5/93.7** | **87.6/93.7** | **85.6/88.7** | **85.7/88.8** | **85.5/88.6** | **86.0/88.9** |

result, we see that AdaLoRA consistently outperforms existing approaches under all the budget levels in term of two evaluation metrics: exact match (EM) and F1. Notice that the performance of Houlsby adapter and Pfeiffer adapter are notably decreased when we reduce the parameter budget. In contrast, our method shows the consistent performance under different budget levels. For example, AdaLoRA achieves 88.7% F1 on SQuADv2.0 with the smallest budget 0.08%. It is close to its performance under the high budget and it is also 1.2% higher than the best-performing baseline.

## 4.3 Natural Language Generation

Table 3: Results with BART-large on XSum and CNN/DailyMail. Here *# Params* is the number of trainable parameters relative to that in full fine-tuning. We report R-1/2/L. The best results are shown in **bold**.

| # Params | Method | XSum | CNN/DailyMail |
|---|---|---|---|
| **100%** | Full FT | **45.49 / 22.33 / 37.26** | 44.16 / 21.28 / 40.90 |
| **2.20%** | LoRA | 43.95 / 20.72 / 35.68 | **45.03** / 21.84 / 42.15 |
| | AdaLoRA | **44.72 / 21.46 / 36.46** | 45.00 / **21.89 / 42.16** |
| **1.10%** | LoRA | 43.40 / 20.20 / 35.20 | 44.72 / 21.58 / 41.84 |
| | AdaLoRA | **44.35 / 21.13 / 36.13** | **44.96 / 21.77 / 42.09** |
| **0.26%** | LoRA | 43.18 / 19.89 / 34.92 | 43.95 / 20.91 / 40.98 |
| | AdaLoRA | **43.55 / 20.17 / 35.20** | **44.39 / 21.28 / 41.50** |
| **0.13%** | LoRA | 42.81 / 19.68 / 34.73 | 43.68 / 20.63 / 40.71 |
| | AdaLoRA | **43.29 / 19.95 / 35.04** | **43.94 / 20.83 / 40.96** |

**Models and Datasets.** To provide a comparison with the state-of-the-art in natural language generation (NLG) tasks, we apply AdaLoRA to fine-tune a BART-large model (Lewis et al., 2019). We evaluate model performance on two datasets: XSum (Narayan et al., 2018) and CNN/DailyMail (Hermann et al., 2015).

**Implementation Details.** Similarly as DeBERTav3-base, we apply low-rank/SVD-based adaptation to every weight matrix of both encoder and decoder layers. We report ROUGE 1/2/L scores (R-1/2/L, Lin (2004)). We set the training epochs as 15. For XSum, we set the beam length as 8 and batch size as 64. For CNN/DailyMail, we set the beam length as 4 and batch size as 32. Please see Appendix G for the detailed configuration.

**Main Results.** Experimental results are summarized in Table 3, where we compare the fine-tuning performance under four budget levels: the number of trainable parameters is 0.13%, 0.26%, 1.10% and 2.20% of total pre-trained parameters. We see that AdaLoRA achieves better or on par performance compared with the baseline on both datasets (XSum and CNN/DailyMail) under all the budget levels. For example, AdaLoRA achieves 21.13 R-2 score when budget level is 1.10%, compared with 19.89 for LoRA.

## 4.4 Analysis

**Different budget levels.** Figure 2 illustrates experimental results of fine-tuning DeBERTaV3-base under different budget levels. We see that on all the three datasets (MNLI-m, SQuADv2.0 and XSum), AdaLoRA achieves consistent performance improvement under all the budget levels compared with the baseline. The performance gain is more significant when increasing the budget for the XSum

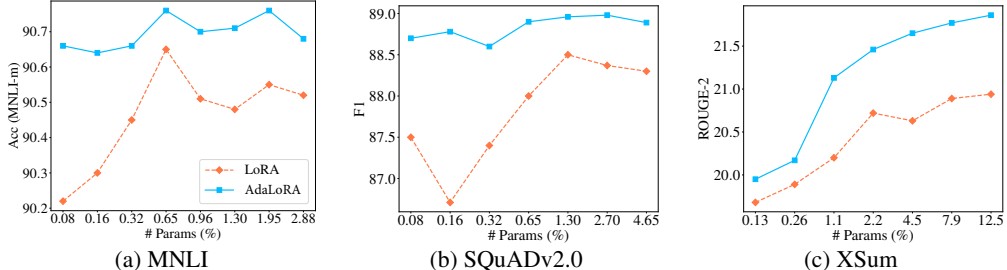

Figure 2: Fine-tuning performance under different budget levels. We compare AdaLoRA with the generalized LoRA that applies to every weight matrix.

task, suggesting a high budget can help NLG tasks. Note that on the MNLI and SQuADv2.0 datasets, the performance of AdaLoRA under low budget levels ($\leq 1\%$) can match the results of high budget settings. For example, AdaLoRA achieves $88.78\%$ F1 on SQuADv2.0 when the budget is $0.16\%$. It is close to the performance ($88.89\%$ F1) of the highest budget ($4.65\%$) with a more significant gain over the baseline.

**Comparison to low-rank parameterization.** As mentioned in Section 3.1, one can alternatively prune LoRA doublet-wise to conduct the rank allocation. In this case, the doublets are zeroed out entirely, raising the barrier to reactivate them. It can cause training instability and hurt the generalization when some crucial doublets are pruned by mistake. In Table 4, we compare AdaLoRA with pruning LoRA on three datasets (SST-2, RTE, and CoLA) to illustrate this point. We apply the same importance score, budget scheduler and training setups as Section 4.1 for pruning LoRA. We can see that AdaLoRA outperforms pruning LoRA on all the datasets under all the budget levels.

Table 4: We present two ablation studies in this table: (i) Comparison between AdaLoRA and structured pruning on LoRA. (ii) Comparison of different importance metrics for AdaLoRA.

| | SST-2 | | | RTE | | | CoLA | | |
|---|---|---|---|---|---|---|---|---|---|
| # Params | 0.08% | 0.16% | 0.65% | 0.08% | 0.16% | 0.65% | 0.08% | 0.16% | 0.65% |
| Prune LoRA | 94.84 | 94.50 | 94.95 | 86.28 | 86.15 | 87.00 | 66.71 | 69.29 | 69.57 |
| AdaLoRA | 95.52 | 95.80 | 96.10 | 87.36 | 87.73 | 88.09 | 70.21 | 70.04 | 71.45 |
| $s(\cdot) = I(\cdot)$ | 94.61 | 95.30 | 95.64 | 87.36 | 87.71 | 88.10 | 66.71 | 68.83 | 70.19 |
| $S_i = |\lambda_i|$ | 95.41 | 95.41 | 95.87 | 87.00 | 86.28 | 88.00 | 67.67 | 68.44 | 70.38 |

**Variants of the importance score.** Recall that in AdaLoRA, the importance score is defined by the sensitivity and uncertainty of every entry in the triplet (7). In Table 4, we examine two variants of the importance score: (i) changing $s(\cdot)$ in (7) to sensitivity-only; (ii) directly defining $S_i$ as $|\lambda_i|$. From the results, we can see that the proposed importance score generally performs best. The other two variants can degenerate the model performance up to $0.9\%$.

**The resulting budget distribution.** Figure 3 in Appendix C shows the resulting rank of each incremental matrix of DeBERTaV3-base fine-tuned with AdaLoRA. We find that AdaLoRA always prefers to allocating more budget to FFNs and top layers. Such behavior aligns with our empirical conclusions presented in Figure 1 that weight matrices of FFN moduels and top layers are more important for model performance. Hence, it validates that our proposed importance metric can guide AdaLoRA to focus on crucial modules. Meanwhile, the rank distribution generated by AdaLoRA is consistent across different budget levels, tasks and models. It means the number of remaining parameters is linearly scaled with $b^{(T)}$ and hence we can tune $b^{(T)}$ to control the remaining parameters.

## 5 CONCLUSION

We propose a parameter-efficient fine-tuning method – AdaLoRA that adaptively allocates the parameter budget according to importance scoring. In AdaLoRA, we parameterize the incremental updates of weight matrices in the form of singular value decomposition. Then, we dynamically allocate the parameter budget among incremental matrices by manipulating the singular values based on a new importance measurement. Such an a pproach effectively improves the model performance and parameter efficiency. We conduct extensive experiments on natural language processing, question answering and natural language generation tasks. Results show that AdaLoRA outperforms existing approaches.

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

## A    THE DETAILED ALGORITHM

---

**Algorithm 1** AdaLoRA

---

1: **Input:** Dataset $\mathcal{D}$; total iterations $T$; budget schedule $\{b^{(t)}\}_{t=0}^T$; hyperparameters $\eta, \gamma, \beta_1, \beta_2$.
2: **for** $t = 1, \ldots, T$ **do**
3:    Sample a mini-batch from $\mathcal{D}$ and compute the gradient $\nabla\mathcal{L}(\mathcal{P}, \mathcal{E}, \mathcal{Q})$;
4:    Compute the sensitivity $I^{(t)}$ in (8) for every parameter in $\{\mathcal{P}, \mathcal{E}, \mathcal{Q}\}$;
5:    Update $\overline{I}^{(t)}$ as (9) and $\overline{U}^{(t)}$ as (10) for every parameter in $\{\mathcal{P}, \mathcal{E}, \mathcal{Q}\}$;
6:    Compute $S_{k,i}^{(t)}$ by (7), for $k = 1, \ldots, n$ and $i = 1, \ldots, r$ ;
7:    Update $P_k^{(t+1)} = P_k^{(t)} - \eta\nabla_{P_k}\mathcal{L}(\mathcal{P}, \mathcal{E}, \mathcal{Q})$ and $Q_k^{(t+1)} = Q_k^{(t)} - \eta\nabla_{Q_k}\mathcal{L}(\mathcal{P}, \mathcal{E}, \mathcal{Q})$;
8:    Update $\Lambda_k^{(t+1)} = \mathcal{T}(\Lambda_k^{(t)} - \eta\nabla_{\Lambda_k}\mathcal{L}(\mathcal{P}, \mathcal{E}, \mathcal{Q}), S_k^{(t)})$ given the budget $b^{(t)}$.
9: **end for**
10: **Output:**

---

## B    GLOBAL BUDGET SCHEDULE

As mentioned in Section 3.3, we propose a global budget scheduler to gradually decrease the budget $b^{(t)}$ following a cubic schedule. The detailed equation is given as follows:

$$b^{(t)} = \begin{cases} b^{(0)} & 0 \leq t < t_i \\ b^{(T)} + \left(b^{(0)} - b^{(T)}\right)\left(1 - \frac{t - t_i - t_f}{T - t_i - t_f}\right)^3 & t_i \leq t < T - t_f \\ b^{(T)} & \text{o.w.} \end{cases} \qquad (12)$$

## C    THE BUDGET DISTRIBUTION

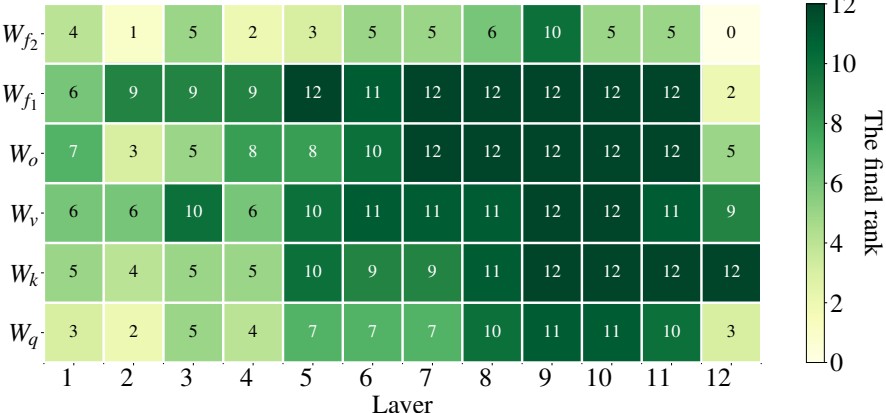

Figure 3: The resulting rank of each incremental matrix when fine-tuning DeBERTaV3-base on MNLI with AdaLoRA. Here the $x$-axis is the layer and the $y$-axis represents different types of adapted weight matrices.

Figure 3 shows the resulting rank of each incremenal matrix when fine-tuning DeBERTaV3-base on MNLI with AdaLoRA. We can see that AdaLoRA allocates more parameter budget to weight matrices of 8, 9, 10, and 11 layers, especially for $W_{f_1}$ and $W_v$. Such behavior aligns with our conclusions presented in Figure 1.

## D    GLUE DATASET STATISTICS

We present the dataset statistics of GLUE (Wang et al., 2019) in the following table.

Table 5: Summary of the GLUE benchmark.

| **Corpus** | Task | #Train | #Dev | #Test | #Label | Metrics |
|---|---|---|---|---|---|---|
| | | | Single-Sentence Classification (GLUE) | | | |
| CoLA | Acceptability | 8.5k | 1k | 1k | 2 | Matthews corr |
| SST | Sentiment | 67k | 872 | 1.8k | 2 | Accuracy |
| | | | Pairwise Text Classification (GLUE) | | | |
| MNLI | NLI | 393k | 20k | 20k | 3 | Accuracy |
| RTE | NLI | 2.5k | 276 | 3k | 2 | Accuracy |
| QQP | Paraphrase | 364k | 40k | 391k | 2 | Accuracy/F1 |
| MRPC | Paraphrase | 3.7k | 408 | 1.7k | 2 | Accuracy/F1 |
| QNLI | QA/NLI | 108k | 5.7k | 5.7k | 2 | Accuracy |
| | | | Text Similarity (GLUE) | | | |
| STS-B | Similarity | 7k | 1.5k | 1.4k | 1 | Pearson/Spearman corr |

# E  NATURAL LANGUAGE UNDERSTANDING

## E.1  BUDGET CONFIGURATION

For each budget level, we tune the final budget $b^{(T)}$ for AdaLoRA, the rank $r$ for LoRA, the hidden dimension $d$ for two adapters to match the budget requirements.

Table 6: Detailed budget setup for GLUE benchmark.

| # Params | Houlsby Adapter ($d$) | Pfeiffer Adapter ($d$) | LoRA ($r$) | AdaLoRA ($b^{(T)}$) |
|---|---|---|---|---|
| 1.2M | 32 | 64 | 8 | 576 |
| 0.6M | 16 | 32 | 4 | 288 |
| 0.3M | 8 | 16 | 2 | 144 |

Alternatively, we can also set the final average rank $\bar{r}^{(T)} = b^{(T)}/n$ for AdaLoRA to control the budget, which is set as 2, 4, and 8 given the final budget as 144, 288, and 576 respectively. Then we select the initial rank $r$ from $\{4, 6, 12\}$ for the final average rank $\{2, 4, 8\}$ respectively.

## E.2  TRAINING DETAILS

We tune the learning rate from $\{8 \times 10^{-5}, 5 \times 10^{-5}, 3 \times 10^{-5}, 1 \times 10^{-4}, 3 \times 10^{-4}, 5 \times 10^{-4}, 8 \times 10^{-4}, 1 \times 10^{-3}\}$ and pick the best learning rate for every method. For each dataset, the batch size is set as identical for every method.

Table 7: Hyper-parameter setup of AdaLoRA for GLUE benchmark.

| Dataset | learning rate | batch size | # epochs | $\gamma$ | $t_i$ | $\Delta_T$ | $t_f$ |
|---|---|---|---|---|---|---|---|
| **MNLI** | $5 \times 10^{-4}$ | 32 | 7 | 0.1 | 8000 | 100 | 50000 |
| **RTE** | $1.2 \times 10^{-3}$ | 32 | 50 | 0.3 | 600 | 1 | 1800 |
| **QNLI** | $1.2 \times 10^{-3}$ | 32 | 5 | 0.1 | 2000 | 100 | 8000 |
| **MRPC** | $1 \times 10^{-3}$ | 32 | 30 | 0.1 | 600 | 1 | 1800 |
| **QQP** | $5 \times 10^{-4}$ | 32 | 5 | 0.1 | 8000 | 100 | 25000 |
| **SST-2** | $8 \times 10^{-4}$ | 32 | 24 | 0.1 | 6000 | 100 | 22000 |
| **CoLA** | $5 \times 10^{-4}$ | 32 | 25 | 0.5 | 800 | 10 | 3500 |
| **STS-B** | $2.2 \times 10^{-3}$ | 32 | 25 | 0.1 | 800 | 10 | 2000 |

# F  QUESTION ANSWERING

## F.1  BUDGET CONFIGURATION

Given the budget, we control the trainable parameters for each method as the following table.

Table 8: Detailed budget setup for question answering.

| # Params | Houlsby Adapter $d$ | Pfeiffer Adapter $d$ | LoRA $r$ | AdaLoRA $b^{(T)}/\bar{r}^{(T)}/r$ |
|---|---|---|---|---|
| 0.65% | 32 | 64 | 8 | 576 / 8 / 12 |
| 0.32% | 16 | 32 | 4 | 288 / 4 / 6 |
| 0.16% | 8 | 16 | 2 | 144 / 2 / 4 |
| 0.08% | 4 | 8 | 1 | 72 / 1 / 2 |

### F.2 TRAINING DETAILS

We set the batch size as 16. We select the learning rate from $\{8 \times 10^{-5}, 5 \times 10^{-5}, 3 \times 10^{-5}, 1 \times 10^{-4}, 3 \times 10^{-4}, 5 \times 10^{-4}, 8 \times 10^{-4}, 1 \times 10^{-3}\}$ and pick the best-performing learning rate for every method. The configuration of AdaLoRA is listed in the following table.

Table 9: Hyper-parameter setup of AdaLoRA for question answering tasks.

| Dataset | learning rate | batch size | # epochs | $\gamma$ | $t_i$ | $\Delta_T$ | $t_f$ |
|---|---|---|---|---|---|---|---|
| **SQuADv1.1** | $1 \times 10^{-3}$ | 16 | 10 | 0.1 | 5000 | 100 | 25000 |
| **SQuADv2.0** | $1 \times 10^{-3}$ | 16 | 12 | 0.1 | 5000 | 100 | 50000 |

### F.3 DATASET

The statistics of question answering datasets are summarized in Table 10.

Table 10: Statistics of the SQuAD dataset.

| | # Train | # Validation |
|---|---|---|
| SQuAD v1.1 | 87,599 | 10,570 |
| SQuAD v2.0 | 130,319 | 11,873 |

## G NATURAL LANGUAGE GENERATION

### G.1 BUDGET CONFIGURATION

Given the budget, we control the trainable parameters for each method as the following table.

Table 11: Detailed budget setup for summarization tasks.

| # Params | Houlsby Adapter $d$ | Pfeiffer Adapter $d$ | LoRA $r$ | AdaLoRA $b^{(T)}/\bar{r}^{(T)}/r$ |
|---|---|---|---|---|
| 0.65% | 32 | 64 | 8 | 576 / 8 / 12 |
| 0.32% | 16 | 32 | 4 | 288 / 4 / 6 |
| 0.16% | 8 | 16 | 2 | 144 / 2 / 4 |
| 0.08% | 4 | 8 | 1 | 72 / 1 / 2 |

### G.2 TRAINING DETAILS

We set the batch size as 16. We select the learning rate from $\{8 \times 10^{-5}, 5 \times 10^{-5}, 3 \times 10^{-5}, 1 \times 10^{-4}, 3 \times 10^{-4}, 5 \times 10^{-4}, 8 \times 10^{-4}, 1 \times 10^{-3}\}$ and pick the best-performing learning rate for every method. The configuration of AdaLoRA is listed in the following table.

Table 12: Hyper-parameter setup of AdaLoRA for summarization tasks.

| Dataset | learning rate | batch size | # epochs | $\gamma$ | $t_i$ | $\Delta_T$ | $t_f$ |
|---|---|---|---|---|---|---|---|
| **XSum** | $5 \times 10^{-4}$ | 64 | 25 | 0.1 | 6000 | 100 | 50000 |
| **CNN/DailyMail** | $5 \times 10^{-4}$ | 32 | 15 | 0.1 | 5000 | 100 | 85000 |

## H  ABLATION STUDY FOR LoRA

As mentioned in Section 4, we find that the performance of LoRA can be further improved when applying it to every weight matrix, compared to fine-tuning $W_q$ and $W_v$ only (Hu et al., 2022). This observation aligns with the empirical results of He et al. (2022). In Table 13, we follow the same training configuration as Section 4.1 and present an ablation study to illustrate this point.

Table 13: We compare the fine-tuning performance when apply LoRA to every weight matrix or $W_q, W_v$ only. The parameter budget is fixed as 0.3M. We report accuracy for QQP and MRPC, accuracy(m) for MNLI, and average correlation for STS-B.

| | MNLI | QQP | CoLA | RTE | QNLI | SST-2 | MRPC | STS-B |
|---|---|---|---|---|---|---|---|---|
| LoRA $(W_q, W_k)$ | 89.80 | 90.48 | 67.04 | 83.75 | 93.69 | 94.84 | 90.20 | 91.05 |
| LoRA (all) | 90.30 | 91.61 | 68.71 | 85.56 | 94.31 | 94.95 | 90.44 | 91.68 |

## I  ORTHOGONAL REGULARIZATION

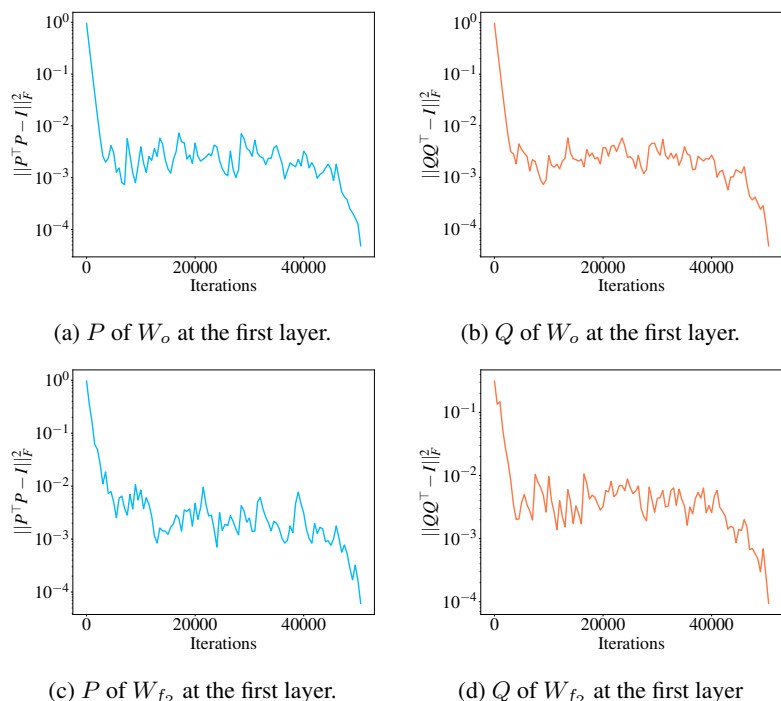

(a) $P$ of $W_o$ at the first layer.

(b) $Q$ of $W_o$ at the first layer.

(c) $P$ of $W_{f_2}$ at the first layer.

(d) $Q$ of $W_{f_2}$ at the first layer

Figure 4: We plot the $\|P^\top P - I\|_F^2$ and $\|QQ^\top - I\|_F^2$ when fine-tuning DeBERTaV3-base on SST-2.

To verify the effectiveness of (4), we plot $\|P^\top P - I\|_F^2$ and $\|QQ^\top - I\|_F^2$ to show whether $P$ and $Q$ are regularized to be orthogonal. We fine-tune a DeBERTaV3-base model on SST-2 with AdaLoRA and follow the same training configuration as Section 4.1. We set $\gamma$ as 0.1 and plot the two terms along the training horizon. From Figure 4, we can see that two regularization terms can be optimized to a very small value (e.g., 0.001) at the beginning of training. Therefore, both $P$ and $Q$ can be

enforced to be orthogonal quickly during the initial warm-up of AdaLoRA. It ensures that the triplets are not dependent with each other.

## J   THE ROLE OF TWO COMPONENTS

We remark that both two components of our method - SVD adaptation and adaptive budget allocation, play vital roles for the performance gain. To demonstrate it, we compare AdaLoRA with the following variants: (i) SVD-LoRA: fine-tuning only with the proposed SVD-based adaptation in (3) and (4); (ii) LoRA$_{\text{regu}}$: LoRA with orthogonal regularization (4) on $A$ and $B$; (iii) AdaLoRA$_{\gamma\,=\,0}$: AdaLoRA without orthogonal regularization (4). Table 14 present the results when fine-tuning DeBERTaVe-base on SST-2 and MNLI. We can see that fine-tuning only with SVD adaptation shows an improvement over LoRA but cannot match the performance of AdaLoRA. Meanwhile, without SVD orthogonal regularization, the performance of AdaLoRA can degenerate. These results validate that both components contribute to the model performance.

Table 14: We present ablation studies about SVD-based adaptation, orthogonal regularization, and budget allocation in this table. For MNLI, we report the average score of m/mm acc.

|  | SST-2 | | | | MNLI | | | |
| --- | --- | --- | --- | --- | --- | --- | --- | --- |
| # Params | 0.08% | 0.16% | 0.32% | 0.65% | 0.08% | 0.16% | 0.32% | 0.65% |
| LoRA | 94.38 | 94.95 | - | 94.95 | 90.19 | 90.34 | - | 90.57 |
| LoRA$_{\text{regu}}$ | - | 94.61 | 94.72 | 94.61 | - | 90.30 | 90.40 | 90.66 |
| SVD-LoRA | 95.33 | 95.18 | 95.07 | 95.53 | 90.28 | 90.25 | 90.52 | 90.62 |
| AdaLoRA$_{\gamma\,=\,0}$ | 95.41 | 95.10 | 95.30 | 95.10 | 90.37 | 90.34 | 90.56 | 90.43 |
| AdaLoRA | 95.64 | 95.80 | 96.10 | 96.10 | 90.65 | 90.68 | 90.66 | 90.77 |

## K   COMPARISON OF TRAINING COST

We compare the training cost between AdaLoRA and LoRA in the following table. We use two methods to fine-tune DeBERTaV3-base on a single NVIDIA V100 GPU. We do training only and set hyperparameters, e.g., batch size and training epochs, the same as in Section 4.

Table 15: Comparison of practical training cost between AdaLoRA and LoRA.

| Dataset | # Param | Method | GPU Mem | Time/epoch |
| --- | --- | --- | --- | --- |
| **MNLI** | **0.08%** | LoRA | 11.094 GB | 105 min |
|  |  | AdaLoRA | 11.104 GB | 116 min |
|  | **0.16%** | LoRA | 11.098 GB | 105 min |
|  |  | AdaLoRA | 11.110 GB | 117 min |
|  | **0.65%** | LoRA | 11.128 GB | 105 min |
|  |  | AdaLoRA | 11.188 GB | 117 min |
| **SST-2** | **0.08%** | LoRA | 13.138 GB | 60 min |
|  |  | AdaLoRA | 13.148 GB | 71 min |
|  | **0.16%** | LoRA | 13.142 GB | 61 min |
|  |  | AdaLoRA | 13.164 GB | 71 min |
|  | **0.65%** | LoRA | 13.170 GB | 61 min |
|  |  | AdaLoRA | 13.226 GB | 71 min |

Table 15 shows that MARVEL incurs 11% additional training time on MNLI and 16% on SQuADv2 under different budgets. The memory footprint of two methods are quite close. Such results demonstrate that MARVEL does not incur significant training overheads. The reason behind is that we only evaluate the importance score for small incremental matrices $P\Lambda Q$. Their total number of parameters is usually less than 1% of pre-trained weights. Therefore, it does not lead to significant computational cost to update the importance scores of these well-structured small matrices, compared to forward-backward pass of full model.

