# OpenReview forum: "Adaptive Budget Allocation for Parameter-Efficient Fine-Tuning "
_ICLR.cc/2023/Conference — ICLR 2023 poster_

### Official Review · Reviewer_z2BR · 2022-10-17

**Confidence:** 3
**Correctness:** 3
**Technical Novelty And Significance:** 3
**Empirical Novelty And Significance:** 3
**Recommendation:** 8

**Clarity, Quality, Novelty And Reproducibility:**

The writing is generally clear (though see above). The intro seems a bit long and could be shortened. E.g., the paragraph starting with "Adding more trainable parameters" is a bit repetitive. Same goes for the background, e.g., no need to formally define transformers, Eq.2 repeats Eq.1, etc.

Typos and such:

- the product of two *samll*

- we present *an* concrete example

In terms of novelty, the proposed method is a novel (as far as I know) combination of known techniques.

**Strength And Weaknesses:**

Strengths:
- A simple and very space-efficient FT method.
- Very strong empirical results.
- The method is quite simple (though there seem to be a lot of important details)

Weaknesses:
- An important ablation seems missing: running LoRA with the proposed SVD decomposition rather than the AB one. This would help tease apart the two proposed contributions (the SVD part and the importance part).

- The writing can be repetitive at times, but still a bit cryptic in other places. In particular, I had to read the SVD part several times to understand the differences between the method and the LoRA method. I still have a few questions, see below.
- The authors only report validation results as far as I understand. Reporting such results is important for reproducibility, but does not replace reporting test results. Given the large number of small decisions made in developing the proposed method, it is particularly important to evaluate it on a truly held-out set.

Questions:
- How much does the initial stage (Sec. 3.3) increase the cost of fine-tuning (e.g., in runtime)?

- "All the gains have passed significant tests with p < 0.05." -> Which statistical test was used?

- "such that the pruned triplets can still get updated within these intervals and possibly reactivated in futur iterations." -> maybe I am missing something: once the warmup stage is over, and the budget is fixed, do importance scores change? In my understanding, the importance of a given singular value cannot change since it is always pruned. Or am I missing something?

- Table 1: Why does the bitfit baseline use far fewer parameters than all the others?

- What are the NLG tasks? Summarization? What are the evaluation measures? These are very important details, NLG is not a task.

**Summary Of The Paper:**

This paper proposes a space-efficient method for finetuning large LMs. The proposed method is a simple extension of the LoRA method: (1) instead of decomposing the update matrix into general A*B, the authors use SVD matrices, which allows reducing the matrix size rather partially than all or none. (2) They propose an importance ranking method for deciding which SVD values to prune in which matrix. The overall method obtains very strong results: it outperforms the LoRA method, as well as several baselines, and even the baseline vanilla fine-tuning.


**Summary Of The Review:**

A simple yet effective extension of the LoRA method for space-efficient fine-tuning of LMs. The proposed method performs very well on several benchmarks. There are a few clarity issues and some methodological concerns, such as an important missing baseline. Overall I am supportive of accepting this paper.

---

> ### Author Response · Authors · 2022-11-18
> **The response to the other questions.**
>
> **Q4: Once the warmup stage is over, and the budget is fixed, do importance scores change? The importance of a given singular value cannot change since it is always pruned?**
>
> Yes, your understanding is correct. After the rank-allocation stage (the second case of Eq. (12), Appendix A), the rank of each incremental matrix is fixed and their importance scores will not be updated. The statement is for the rank allocation stage. That is, we mask out singular values of unimportant triplets every $\Delta_T$ steps during the rank-allocation stage. Between the interval of two masking steps, we update $P, \Lambda, Q$ as usual so that we can recover the crucial triplets dropped by mistake.
>
> **Q5: Other questions.**
>
> + Why does the bitfit baseline use far fewer parameters than all the others?
>
> It is because Bitfit only fine-tunes the bias terms of each linear module. Hence the total number of trainable parameters is very small.
>
> + Which statistical test was used?
>
> Sorry for the typos. It should be “significance test”. We use the two-sample t test.
>
> + What are the NLG tasks? Summarization?
>
> Yes, it means the summarization tasks. It generates a summarization from the input text. We compare the generated summarization to the reference sentence and report the rouge score (R-1/2/L).
>
> **Q6: Test accuray.**
>
> As our baseline LoRA has not reported their test accuracy, we did not find the reference to compare with. Hence we just follow the common practice to report the validation accuracy for all methods. We will include the test accuracy in the future version.

---

> > ### Comment · Reviewer_z2BR · 2022-11-22
> > **Thank you for the detailed response**
> >
> > I am satisfied with these responses. I am looking forward to seeing these clarifications (and those promised) in the final version.

---

> ### Author Response · Authors · 2022-11-18
> **Thanks for your constructive feedback.**
>
> We appreciate your constructive feedback. We provide our response to your questions as follows.
>
> **Q1: An important ablation seems missing: running LoRA with the proposed SVD decomposition rather than the AB one.**
>
> We present the ablation study required by the reviewer. We compare MARVEL with the following variants when fine-tuning DeBERTaV3-base on MNLI and SST-2:
>
> + LoRA-regu: LoRA with orthogonal regularization on $A, B$.
> + SVD LoRA: Fine-tune models with the proposed SVD adaptation (Eq. (3) and (4)).
> + MARVEL with regularization coefficient $\gamma = 0$.
>
> |   |   | SST-2  |   |   |
> | :-----: | :-----: | :-----: | :-----: | :-----: |
> | **# Params** | **0.08%** | **0.16%** | **0.32%** | **0.65%** |
> | LoRA | 94.38 | 94.95 |  -  | 94.95 |
> | LoRA-regu |     -     | 94.61 | 94.72 | 94.61 |
> | SVD LoRA     | 95.33 | 95.18 | 95.07 | 95.53 |
> | MARVEL($\gamma=0$) | 95.41  |  95.10 | 95.30  | 95.10 |
> | MARVEL        | 95.64 | 95.80 | 96.10 | 96.10 |
> |    |   | **MNLI** (m/mm) |   |   |
> | **# Params** | **0.08%** | **0.16%** | **0.32%** | **0.65%** |
> | LoRA | 90.20/90.17 | 90.30/90.38 |  - / -  | 90.65/90.69 |
> | LoRA-regu |       - / -         | 90.24/90.35 | 90.35/90.44 | 90.63/90.70 |
> | SVD LoRA     | 90.18/90.38 | 90.14/90.37 | 90.44/90.60 | 90.60/90.64 |
> | MARVEL($\gamma=0$) | 90.33/90.40  | 90.30/90.39  | 90.52/90.60  | 90.35/90.52  |
> | MARVEL       | 90.64/90.65 | 90.66/90.70 | 90.66/90.65 | 90.76/90.79 |
>
> From the table above, we can see that both two components - SVD adaptation and adaptive rank allocation, are necessary for the performance gains. Fine-tuning only with SVD adaptation shows an improvement over LoRA but cannot match the performance of MARVEL. Meanwhile, without SVD orthogonal regularization, the performance of MARVEL can degenerate. These results validate that both components contribute to the model performance.
>
> **Q2: The writing can be repetitive at times, but still a bit cryptic in other places.**
>
> Thanks for your helpful suggestions. We will revise the paper as suggested, especially for the SVD-based adaptation part.
>
> **Q3: How much does the initial stage (Sec. 3.3) increase the cost of fine-tuning (e.g., in runtime)?**
>
> We run MARVEL and LoRA for the same number of training epochs. We usually set the initial warm-up steps $t_i$ (Eq.(12) in Appendix A) as the 1/10 of total training steps and the final warm-up step $t_f$ as 1/2 of total training steps. The runtime is mainly increased by iteratively updating importance scores and pruning unimportant triplets. To compare the practical training time, we present the following table. We use MARVEL and LoRA to fine-tune DeBERTaV3-base on a single NVIDIA V100 GPU. We do training only and set hyperparameters as the same as Section 4 (See Appendix D and E).
>
> | Dataset | # Params | Method | GPU Mem | Time/epoch |
> | :----- | :-----: | :-----: | :-----: | :-----: |
> | MNLI | 0.08% | LoRA | 11.094GB | 105min |
> | MNLI | 0.08% | MARVEL | 11.104GB | 116min |
> | MNLI | 0.16% | LoRA | 11.098GB | 105min |
> | MNLI | 0.16% | MARVEL | 11.110GB | 117min |
> | MNLI | 0.65% | LoRA | 11.128GB | 105min |
> | MNLI | 0.65% | MARVEL | 11.188GB | 117min |
> | SQuADv2 | 0.08% | LoRA | 13.138GB  | 60min |
> | SQuADv2 | 0.08% | MARVEL | 13.148GB | 71min |
> | SQuADv2 | 0.16% | LoRA | 13.142GB | 61min |
> | SQuADv2 | 0.16% | MARVEL | 13.164GB  | 71min |
> | SQuADv2 | 0.65% | LoRA | 13.170GB | 61min |
> | SQuADv2 | 0.65% | MARVEL | 13.226GB  | 71min |
>
> As iteratively updating the importance scores, MARVEL incurs 11% additional training time on MNLI and 16% on SQuADv2. The memory footprint of two methods are quite close. The results suggest that MARVEL incurs additional but not significant training overhead whereas MARVEL achieves significant performance gains on various tasks.

---

### Official Review · Reviewer_97di · 2022-10-25

**Confidence:** 3
**Correctness:** 2
**Technical Novelty And Significance:** 2
**Empirical Novelty And Significance:** 3
**Recommendation:** 6

**Clarity, Quality, Novelty And Reproducibility:**

- Not all parameters take the same amount of FLOPS for updates. For example, computing the gradient of weights in the upper layers would be faster than computing those in the bottom layers (due to backpropagation). It's unclear how the proposed method speeds up the training in terms of FLOPS (or training time) compared to other methods.
- The main purpose of efficient tuning is to speed up the fine-tuning process and maintain comparable performance to full model tuning. Note that **controlling model parameters is just one way to do so but not the main target**. In Algorithm 1, we have to first update P_k and Q_k (Line 7), and then take a gradient update to get the diagonal matrix (Eq. 5), and finally prune by Eq. (6). Computing the gradient updates is the most time-consuming part. If we have already spent time computing the gradients of P_k, Q_k, \Lambda_k (Line 7-8), **what's the point of pruning?** Note that just controlling the no. of fine-tuning parameters is not the main focus, it does not justify the training efficiency purpose. The overhead of computing the sensitivity score (Line 4-6) and pruning (Eq. 6) may just make the whole training process slower compared to the training process only doing the gradient updates of P_k, Q_k, \Lambda_k.
- It is claimed that one of the advantages of MARVAL is to maintain the singular vectors for potential future recovery. In which situation should we recover these triplets? There is no support for this statement in the experiment.



**Strength And Weaknesses:**

Strength:
- The paper is well-written and easy to follow. The research problem is important as pre-training LMs are growing in scale, resulting in the difficulty in fine-tuning all model parameters.
- The method is simple, straightforward, and effective.

Weaknesses:
- This paper mainly compares methods under the same amount of fine-tuning parameters. There is no comparison in terms of FLOPS/training time.
- Besides, it's also unclear about the speedup gain v.s. the overhead of computing the sensitivity score w.r.t. the whole training process.

**Summary Of The Paper:**

This paper presents a parameter-efficient fine-tuning method for pre-trained language models. The method first approximates the gradient update of weight matrics by low-rank factorization and uses a sensitivity-based importance score to prune unimportant triplets. Experiments on NLU, QA, NLG tasks show the effectiveness of the proposed method compared to other methods with the same amount of fine-tuning parameters.

**Summary Of The Review:**

The motivation for adjusting the rank of weight matrics is not very clear. Experimental results in terms of FLOPS (or training time) are needed to show the efficiency of the proposed method.

------------------
The score is updated after the responses.

---

> ### Author Response · Authors · 2022-11-18
> **The response to the other questions.**
>
> **Q2:  If we have already spent time computing the gradients of $P_k, Q_k, \Lambda_k$, what's the point of pruning? Note that just controlling the number of fine-tuning parameters is not the main focus, it does not justify the training efficiency purpose. The overhead of computing the sensitivity and pruning may make the training slower compared to only doing the gradient updates.**
>
> We would remark again that the main purpose of parameter-efficient fine-tuning is to increase the parameter efficiency when deploying PLMs on a large number of tasks. Given this motivation, we adaptively allocate the parameter budget among modules and layers according to their importance scores to improve model performance and efficiency. The trade-off of achieving this goal is to evaluate importance scores of small incremental matrices $P_k, \Lambda_k, Q_k$. It can induce more training overheads but they are acceptable as the total parameters of all $P_k, \Lambda_k, Q_k$ is usually less than 1% of pre-trained parameters. Updating their importance scores does not incur significant computation cost, compared to full-model forward-backward pass. However, MARVEL can achieve superior model performance with extremely low parameter budget. For example, with 0.08% parameters, MARVEL achieves 1.1% EM gain, compared to that of LoRA with 4 times of parameter budget (0.32%) (See Table 2). Therefore, such a trade-off is worthwhile. We present the training time comparison in the next question.
>
> **Q3: There is no comparison in terms of FLOPS/training time.**
>
> We compare the practical training cost between MARVEL and LoRA in the following table. We use two methods to fine-tune DeBERTaV3-base on single NVIDIA V100 GPU. We do training only and set hyperparameters as the same as Section 4 (See Appendix D and E).
>
> | Dataset | # Params | Method | GPU Mem | Time/epoch |
> | :----- | :-----: | :-----: | :-----: | :-----: |
> | MNLI | 0.08% | LoRA | 11.094GB | 105min |
> | MNLI | 0.08% | MARVEL | 11.104GB | 116min |
> | MNLI | 0.16% | LoRA | 11.098GB | 105min |
> | MNLI | 0.16% | MARVEL | 11.110GB | 117min |
> | MNLI | 0.65% | LoRA | 11.128GB | 105min |
> | MNLI | 0.65% | MARVEL | 11.188GB | 117min |
> | SQuADv2 | 0.08% | LoRA | 13.138GB  | 60min |
> | SQuADv2 | 0.08% | MARVEL | 13.148GB | 71min |
> | SQuADv2 | 0.16% | LoRA | 13.142GB | 61min |
> | SQuADv2 | 0.16% | MARVEL | 13.164GB  | 71min |
> | SQuADv2 | 0.65% | LoRA | 13.170GB | 61min |
> | SQuADv2 | 0.65% | MARVEL | 13.226GB  | 71min |
>
> As iteratively updating the importance scores, MARVEL incurs 11% additional training time on MNLI and 16% on SQuADv2. The memory footprint of two methods are quite close. The results suggest that MARVEL incurs additional but not significant training overhead whereas it can achieve superior performance on various tasks.
>
> **Q4: It is claimed that one of the advantages of MARVEL is to maintain the singular vectors for potential future recovery. In which situation should we recover these triplets?**
>
> The future recovery is a potential advantage of our method. We will clarify it more in the next version. Basically, as the sensitivity is evaluated on mini batches, it leads to high uncertainty for importance estimation [1]. Some crucial triplets can be evaluated as unimportant on certain training batches. Hence they can be dropped by mistake when getting these batches. However they are still important in general. MARVEL only masks out singular values and maintains singular vectors. Hence the dropped triplets can still be updated based on its existing basis if recovered in future steps. By contrast, structured pruning on LoRA has to mask out entire doublets which makes it harder to recover them as they are all zero and not trained. We present an ablation study in Section 4.4 (Table 4) to illustrate this point. We can see that pruning on LoRA cannot match the performance of MARVEL.
>
> [1] Qingru Zhang, Simiao Zuo, Chen Liang, Alexander Bukharin, Pengcheng He, Weizhu Chen, and Tuo Zhao. Platon: Pruning large transformer models with upper confidence bound of weight importance. ICML 2022.

---

> ### Author Response · Authors · 2022-11-18
> **The motivation of parameter-efficient fine-tuning**
>
> Thanks for your helpful feedback. We provide our response to your questions as follows.
>
> **Q1: The main purpose of efficient tuning is to speed up the fine-tuning process. Note that just controlling the number of fine-tuning parameters is not the main focus, it does not justify the training efficiency purpose.**
>
> We believe there may be some misunderstanding here. As mentioned in Section 1, the major motivation of parameter-efficient fine-tuning is to decrease the memory footprint when serving large pre-trained models (PLM) to a large number of tasks[1,2,3]. The same motivation has also been highlighted by [1,2,3]. That is, we do not have to store a separate copy of full models for each task, which is prohibitively expensive. Instead, parameter-efficient fine-tuning methods, like adapter-tuning, predix-tuning, LoRA, etc., only update a small number of task-specific parameters and store them for each down-steam task. Therefore, the major purpose of these methods is to increase the parameter efficiency when deploying PLMs to multiple tasks [1,2,3].
>
> Moreover, their speedup in terms of training time is limited due to the unavoidable full-model forward pass.  Therefore, the training speedup is not the main purpose of parameter-efficient fine-tuning methods.
>
> [1] Junxian He, Chunting Zhou, Xuezhe Ma, Taylor Berg-Kirkpatrick, and Graham Neubig. Towards a unified view of parameter-efficient transfer learning. ICLR 2022.
>
> [2] Edward J Hu, yelong shen, Phillip Wallis, Zeyuan Allen-Zhu, Yuanzhi Li, Shean Wang, Lu Wang, and Weizhu Chen. LoRA: Low-rank adaptation of large language models. ICLR 2022.
>
> [3] Neil Houlsby, Andrei Giurgiu, Stanislaw Jastrzebski, Bruna Morrone, Quentin De Laroussilhe, Andrea Gesmundo, Mona Attariyan, and Sylvain Gelly. Parameter-efficient transfer learning for nlp. ICML 2019.

---

> > ### Comment · Reviewer_97di · 2022-11-20
> > **Updated score**
> >
> > Thanks for the responses. I've updated my score from 5 to 6, given that the computation overhead is still reasonable and the justification of memory constraint makes sense.

---

### Official Review · Reviewer_qo5b · 2022-10-29

**Confidence:** 4
**Correctness:** 4
**Technical Novelty And Significance:** 3
**Empirical Novelty And Significance:** Not applicable
**Recommendation:** 6

**Clarity, Quality, Novelty And Reproducibility:**

The paper is written in clarity. The method is novel -- it might be a bit niche. The paper seems to have great reproducibility, with detailed hyper-parameters both in the paper and appendices.

**Strength And Weaknesses:**

* Strength
    * The evaluation is extensive. The paper compares with a comprehensive set of baselines and evaluates on a variety of tasks (language generation, GLUE, and question answering).
    * The paper achieves great experimental results, e.g. achieving 1.2% F1 improvement on Squad2. With less than 0.1% trainable parameters.
    * This paper proposes a novel approach for parameter-efficient finetuning, and can have wide applications.
* Weakness
    * It will be great to also provide numbers of training/inference memory/time. Even though it's faster than SVD, is the training time longer than other methods? What are some trade-offs?


**Summary Of The Paper:**

* This paper proposes a novel parameter-efficient fine-tuning method called MARVEL, which adaptively allocates the parameter budget among weight matrices according to their importance score. Specifically, it proposes to use a learned approximation of SVD decomposition, and prunes out singular values of unimportant updates. Learned approximation of SVD circumvent intensive exact SVD computations, and importance of singular values is defined by a novel metric correlating to the contribution to the model performance. It additionally proposes a global budget scheduler to facilitate training process.


**Summary Of The Review:**

This paper presents a novel method that achieves better efficiency without much loss of accuracy, compared to a comprehensive set of baselines, on a variety of tasks. However, the novelty of the method is somewhat limited and there can be potential drawbacks such as longer training time. So I recommend weak acceptance.

---

> ### Author Response · Authors · 2022-11-18
> **We appreciate your feedback.**
>
> Thanks for your helpful feedback. We provide the response to your questions as follows.
>
> **Q1: Even though MARVEL is faster than SVD, is the training time longer than other methods? What are some trade-offs?**
>
> We compare the practical training cost between MARVEL and LoRA in the following table. We use two methods to fine-tune DeBERTaV3-base on single NVIDIA V100 GPU. We do training only and set hyperparameters as the same as Section 4 (See Appendix D and E).
>
> | Dataset | # Params | Method | GPU Mem | Time/epoch |
> | :----- | :-----: | :-----: | :-----: | :-----: |
> | MNLI | 0.08% | LoRA | 11.094GB | 105min |
> | MNLI | 0.08% | MARVEL | 11.104GB | 116min |
> | MNLI | 0.16% | LoRA | 11.098GB | 105min |
> | MNLI | 0.16% | MARVEL | 11.110GB | 117min |
> | MNLI | 0.65% | LoRA | 11.128GB | 105min |
> | MNLI | 0.65% | MARVEL | 11.188GB | 117min |
> | SQuADv2 | 0.08% | LoRA | 13.138GB  | 60min |
> | SQuADv2 | 0.08% | MARVEL | 13.148GB | 71min |
> | SQuADv2 | 0.16% | LoRA | 13.142GB | 61min |
> | SQuADv2 | 0.16% | MARVEL | 13.164GB  | 71min |
> | SQuADv2 | 0.65% | LoRA | 13.170GB | 61min |
> | SQuADv2 | 0.65% | MARVEL | 13.226GB  | 71min |
>
> As iteratively updating the importance scores, MARVEL incurs 11% additional training time on MNLI and 16% on SQuADv2. The memory footprint of two methods are quite close. The results suggest that MARVEL incurs additional but not significant training overhead. That is because we only evaluate the importance score for small incremental matrices $P\Lambda Q$. Their total number of parameters is usually less than 1% of pre-trained parameters. Therefore, it does not incur significant computational cost to update the importance scores of these well-structured small matrices, compared to forward-backward pass of full model. Moreover, the training cost does not scale with the parameter budget.
>
> For the inference cost, the obtained incremental matrix $P\Lambda Q$ can be merged with the pre-trained weight $W^{(0)}$ when doing inference. Therefore, the inference cost is the same as LoRA and full model.

---

### Official Review · Reviewer_byZf · 2022-10-30

**Confidence:** 3
**Correctness:** 3
**Technical Novelty And Significance:** 2
**Empirical Novelty And Significance:** 4
**Recommendation:** 8

**Clarity, Quality, Novelty And Reproducibility:**

This paper is clear. The quality is very high. The technical contribution is incremental, but the empirical contributions are significant and new. The codebase required to reproduce results are all provided in supplementary materials.

**Strength And Weaknesses:**

## Strengths

1. Parameter-efficient fine-tuning is a topic of significance in the application of big pre-trained LMs to real-world problems.

2. MARVEL uses SVD and dynamically allocate rank to each layer as resources. It is a novel technical contribution.

3. The paper compares MARVEL with multiple baselines on very well-established NLP models (DeBERTa, BART) and benchmarks (GLUE, SQuAD, CNN/DM, XSum). In these experiments, MARVEL outperforms other methods, especially LoRA, consistently.

4. This work includes an analysis section, where various design choices of MARVEL are compared. It helps readers understand how MARVEL works. The figure in Appendix B is also provides great insight on how different layers of big LM contributes to its downstream performance.

## Weaknesses

1. The method, MARVEL, is based on LoRA. Most important comparisons in this paper is comparing MARVEL against LoRA. The contribution of MARVEL is mostly incremental and empirical.

2. MARVEL introduces many new hyperparameters compared with LoRA (see Section 4.1). How could you rule out the possibility that a **larger hyperparameter search space** is what leads to a better performance? Also, MARVEL is considerably more complicated than LoRA, so fine-tuning with MARVEL could be slower than fine-tuning with LoRA. However, the **training time** is not compared in this paper either. To address these two concerns, could you show the time required for each run (including full hyperparameter sweep) of both MARVEL and LoRA?

**Summary Of The Paper:**

This paper proposes MARVEL, a method for parameter-efficient fine-tuning on big pre-trained language models. MARVEL is largely based on LoRA, a method that fine-tunes low-rank decomposition on parameter matrices. MARVEL uses SVD and dynamically allocate rank to each layer as resources, and it outperforms LoRA consistently with both DeBERTa and BART on natural language understanding and natural language generation respectively.

**Summary Of The Review:**

This paper proposes MARVEL, a method for parameter-efficient fine-tuning on big pre-trained language models. The topic is of significance. The method is based on LoRA, so technical contribution is incremental. However, the paper includes very solid experiments showing that MARVEL outperforms LoRA and other baselines consistently. There is a minor soundness problem: some experiments could be unfair against LoRA because training time is not compared. This paper is overall solid so I would like to give a borderline accept recommendation.

---

> ### Author Response · Authors · 2022-11-18
> **Thanks for your helpful feedback.**
>
> Thanks for your constructive feedback. We provide the response to your questions as follows.
>
> **Q1: Most important comparisons in this paper are comparing MARVEL against LoRA. The contribution of MARVEL is mostly incremental and empirical.**
>
> As far as we know, this is the first work to allocate the parameter budget for efficient fine-tuning methods. Alternatively, one can also prune $AB$ of LoRA doublet-wise to allocate the budget. However, such an approach does not significantly outperform LoRA. To showcase that, we present the results from Table 4 and Table 1 as follows:
>
> | Method | SST-2 | SST-2 | RTE | RTE | CoLA | CoLA |
> | :-----: | :-----: | :-----: | :-----: | :-----: | :-----: | :-----: |
> | # Params | 0.16% | 0.65% | 0.16% | 0.65% | 0.16% | 0.65% |
> | LoRA | 94.95 | 94.95 | 85.56 | 85.20 | 68.71 | 69.82 |
> | Prune $AB$ | 94.50 | 94.95 | 86.15 | 87.00 | 69.29 | 69.57 |
> | MARVEL | 95.80 | 96.10 | 87.36 | 88.09 | 70.04 | 71.45 |
>
> We can see that pruning $AB$ performs similarly as LoRA on SST-2 and CoLA. By contrast, MARVEL achieves manifest improvement over both pruning $AB$ and LoRA. This is because the proposed SVD-based adaptation makes it natural to allocate the parameter budget. We can control the singular values to adjust the rank instead of entire doublets. Therefore, the proposed SVD-based adaptation is a novel and nontrivial approach.
>
> **Q2:  How could you rule out the possibility that a larger hyperparameter search space is what leads to a better performance?**
>
> For $\beta_1, \beta_2$, we do not tune them and just use their default values (0.85) as Zhang, et., al., (2022) [1]. For the budget scheduler, we always set the initial budget $b^{(0)}$ as 1.5 times of the final budget $b^{(T)}$ and hardly tune it. For $\gamma$, we just selected from {0.1, 0.3, 0.5}. To further evaluate the hyperparameter sensitivity, we do ablation study in the following table, where we fine-tune DeBERTaV3-base on MNLI with MARVEL.
>
> | # Params | $\gamma$ | $\beta_1$ | $\beta_2$ | MNLI-m |
> | :----- | :-----: | :-----: | :-----: | :-----: |
> | 0.65% | 0.1 | 0.85 | 0.85 | 90.73 |
> | 0.65% | 0.3 | 0.85 | 0.85 | 90.67 |
> | 0.65% | 0.5 | 0.85 | 0.85 | 90.80 |
> | 0.65% | 0.8 | 0.85 | 0.85 | 90.71 |
> | 0.65% | 0.1 | 0.50 | 0.85 | 90.68 |
> | 0.65% | 0.1 | 0.90 | 0.85 | 90.70 |
> | 0.65% | 0.1 | 0.85 | 0.50 | 90.65 |
> | 0.65% | 0.1 | 0.85 | 0.90 | 90.69 |
>
> From the table, we can see that MARVEL is not sensitive to $\beta_1$ and $\beta_2$ and hence we usually set them as the default value $0.85$. For $\gamma$, we usually choose from {0.1, 0.5}.
>
> **Q3: The training time is not compared in this paper.**
>
> We compare the practical training cost between MARVEL and LoRA in the following table. We use two methods to fine-tune DeBERTaV3-base on single NVIDIA V100 GPU. We do training only and set hyperparameters as the same as Section 4 (See Appendix D and E). Specifically, for MNLI, we set the batch size as 32 and training epochs as 5. For SQuADv2, we set the batch size as 16 and training epochs as 12.
>
> | Dataset | # Params | Method | GPU Mem | Time/epoch |
> | :----- | :-----: | :-----: | :-----: | :-----: |
> | MNLI | 0.08% | LoRA | 11.094GB | 105min |
> | MNLI | 0.08% | MARVEL | 11.104GB | 116min |
> | MNLI | 0.16% | LoRA | 11.098GB | 105min |
> | MNLI | 0.16% | MARVEL | 11.110GB | 117min |
> | MNLI | 0.65% | LoRA | 11.128GB | 105min |
> | MNLI | 0.65% | MARVEL | 11.188GB | 117min |
> | SQuADv2 | 0.08% | LoRA | 13.138GB  | 60min |
> | SQuADv2 | 0.08% | MARVEL | 13.148GB | 71min |
> | SQuADv2 | 0.16% | LoRA | 13.142GB | 61min |
> | SQuADv2 | 0.16% | MARVEL | 13.164GB  | 71min |
> | SQuADv2 | 0.65% | LoRA | 13.170GB | 61min |
> | SQuADv2 | 0.65% | MARVEL | 13.226GB  | 71min |
>
> The table shows that MARVEL incurs 11% additional training time on MNLI and 16% on SQuADv2 under different budgets. The memory footprint of two methods are quite close. Such results demonstrate that MARVEL does not incur significant training overheads. That is because we only evaluate the importance score for small incremental matrices $P\Lambda Q$. Their total number of parameters is usually less than 1% of pre-trained parameters. Therefore, it does not incur significant computational overheads to update their importance scores.

---

> > ### Comment · Reviewer_byZf · 2022-11-19
> > **Thank you for your response**
> >
> > Thank you for addressing my concerns on hyperparameter sensitivity and training time. I decide to increase my review score to *8 (accept)*.

---

### Official Review · Reviewer_f1zB · 2022-11-02

**Confidence:** 4
**Correctness:** 3
**Technical Novelty And Significance:** 3
**Empirical Novelty And Significance:** 3
**Recommendation:** 6

**Clarity, Quality, Novelty And Reproducibility:**

*Clarity* :  Method described is clear

*Originality* : Method is original in the context of previous work as far as I can tell but approach  could be considered incremental

*Writing Quality* :   Writing quality can be improved - especially introduction - too many details given in introduction which late feel repeated in later parts of the paper


**Strength And Weaknesses:**

**Strengths**
1. The idea of adaptive computation budget allocation is interesting and highly relevant for using pre-trained models for downstream tasks in a memory efficient fashion
2. The experiments, though limited in some respect, are quite extensive in others.

**Weaknesses**
1. Missing Baselines / Experiments
     1.  MAM [1] seems to be the current state of the art but this was not compared against though it is referenced in the paper.
     2.  What is the training time cost of  Marvel compared to the non-adaptive methods ? It seems that Marvel adds non-trivial overhead compared to the simpler non-adaptive methods.  If the training overhead is significant, it might hinder Marvel's applicability, just like the authors mention with Diff Pruning.
2. Some Missing ablations
    1. Impact of orthogonality constraint - would be good to see ablations with and without this constraint
    2. Is the success of the gradient-weight product (as a sensitivity measure) possibly coming from the exponential smoothing ? Was exponential smoothing applied to the simple singular value magnitude measure (?)
    3.  LoRA version as presented has no regularization applied to the learned AB matrices.  It is possible that the gain produced by the current method is coming from “effective regularization effect"  of the global budget scheduler ?  Could LoRA be run with varying regularization (say L2) on the AB matrices  to ensure that such a regularization effect isn't indeed the cause of gains.


**Nits**
1. Figure 1
    1. We are not giving the reference result for full fine-tuning so it’s hard to put this figure in context
    2. Would be good to also include results for when budget is split equally across all layers / types of parameters

[1] Junxian He, Chunting Zhou, Xuezhe Ma, Taylor Berg-Kirkpatrick, and Graham Neubig. Towards
a unified view of parameter-efficient transfer learning. In International Conference on Learning
Representations, 2022. URL https://openreview.net/forum?id=0RDcd5Axok.

**Summary Of The Paper:**

This paper presents an approach for adaptively allocating fine-tuning parameters during transfer of pre-trained models to downstream tasks.  The authors propose an SVD inspired decomposition of the adapter matrices and develop various importance scores to assess which triplets in the SVD decomposition are removable. This allows adaptively tuning the ranks of the adapter matrices across layers. The authors demonstrate empirical gains from using their method.

**Summary Of The Review:**

I think the ideas in the paper are solid. However, there are several missing empirical elements that I would need to be fully convinced that the method works as advertised (significantly improves downstream performance at same overall budget compared to simpler non-adaptive methods).
Conditioned on adequate responses being made to the weaknesses outlined above, I would be willing to raise my score.

******

Score updated after responses.

---

> ### Author Response · Authors · 2022-11-18
> **More experimental Results**
>
> **Q4: Could LoRA be run with varying regularization on the AB matrices to ensure that such a regularization effect isn't indeed the cause of gains?**
>
> We present more ablation results to verify the effectiveness of two components of MARVEL. Specifically, we compare MARVEL with the following variants when fine-tuning DeBERTaV3-base on MNLI and SST-2:
> + LoRA-regu: LoRA with orthogonal regularization on $A, B$.
> + SVD LoRA: Fine-tune models with the proposed SVD adaptation (Eq. (3) and (4)).
> + MARVEL with regularization coefficient $\gamma = 0$.
>
> |   |   | SST-2  |   |   |
> | :-----: | :-----: | :-----: | :-----: | :-----: |
> | **# Params** | **0.08%** | **0.16%** | **0.32%** | **0.65%** |
> | LoRA | 94.38 | 94.95 |  -  | 94.95 |
> | LoRA-regu |     -     | 94.61 | 94.72 | 94.61 |
> | SVD LoRA     | 95.33 | 95.18 | 95.07 | 95.53 |
> | MARVEL($\gamma=0$) | 95.41  |  95.10 | 95.30  | 95.10 |
> | MARVEL        | 95.64 | 95.80 | 96.10 | 96.10 |
> |    |   | **MNLI** (m/mm) |   |   |
> | **# Params** | **0.08%** | **0.16%** | **0.32%** | **0.65%** |
> | LoRA | 90.20/90.17 | 90.30/90.38 |  - / -  | 90.65/90.69 |
> | LoRA-regu |       - / -         | 90.24/90.35 | 90.35/90.44 | 90.63/90.70 |
> | SVD LoRA     | 90.18/90.38 | 90.14/90.37 | 90.44/90.60 | 90.60/90.64 |
> | MARVEL($\gamma=0$) | 90.33/90.40  | 90.30/90.39  | 90.52/90.60  | 90.35/90.52  |
> | MARVEL       | 90.64/90.65 | 90.66/90.70 | 90.66/90.65 | 90.76/90.79 |
>
> From the table above, we can see that *both* two components - SVD adaptation and adaptive rank allocation, are necessary for the performance gains. Fine-tuning only with SVD adaptation shows an improvement over LoRA but cannot match the performance of MARVEL. Meanwhile, without SVD orthogonal regularization, the performance of MARVEL can degenerate. These results validate that both components contribute to the model performance. Moreover, we also run LoRA with unnormalized orthogonal regularization (LoRA-regu). It means we regularize rows/columns of $A$ / $B$ orthogonal with each other but unnormalized. The result shows such an approach still cannot match the performance of MARVEL.
>
> **Q5: MAM [2] seems to be the current state of the art but this was not compared against.**
>
> Thanks for your helpful suggestion. We add the comparison with MAM Adapter about fine-tuning BART-large on XSum in the following table. We follow the source code and training configuration provided by He, et.,al, (2022) [2] to reproduce the results.
> The MAM adapter contains two components: a scaled parallel adapter (PA) to modify FFNs and a prefix module to generate soft tokens. However, the MAM adapter cannot evaluate module importance nor allocate parameter budget accordingly. We usually tune the size of two modules ***manually*** to allocate parameters appropriately and improve model performance. In the following comparison, we equally split the parameter budget to two modules and compare it with MARVEL.
>
> Specifically, we select the hidden dimension of PA from {5, 10, 42, 84}. For prefix modules, there are two variants - with/without two-layer projections. If there is no linear projection, prefix modules can be regarded as embeddings. Its number of parameters is controlled by the number of generated tokens $l$, which we select from {4, 8, 32, 64}. With the projections, it is mainly dominated by the hidden dimension of the projections, which we select from {3, 6, 24, 48} and fix $l$ as 32. All the other configurations are set identically as provided by He, et.,al, (2022) [2].
>
> |# Param | LoRA | MAM (Prefix w.o. MLP) | MAM (Prefix w. MLP) | MARVEL |
> | :-----: | :-----: | :-----: | :-----: | :-----: |
> | 2.20% | 43.95/20.72/35.68 | 44.06/20.94/36.24 | 43.97/20.87/36.15 | 44.72/21.46/36.46 |
> | 1.10% | 43.40/20.20/35.20 | 43.39/20.39/35.51 | 43.39/20.43/35.75 | 44.35/21.13/36.13 |
> | 0.26% | 43.18/19.89/34.92 | 42.17/19.24/34.33 | 42.12/19.08/34.13 | 43.55/20.17/35.20 |
> | 0.13% | 42.81/19.68/34.73 | 41.67/18.69/33.81 | 41.47/18.45/33.48 | 43.29/19.95/35.04 |
>
> In the table above, we report R1/2/L. We can see that MAM adapter shows slight improvements over LoRA when the budget is high (e.g., 2.20% and 1.10%) but its performance degenerates in the low-budget settings. MARVEL outperforms both LoRA and the MAM adapter under all budget levels.
>
> **Q6: Questions about Figure 1.**
>
> Thanks for your suggestions. The results of full fine-tuning and equally splitting budget can be found in Table 1. We will add these results to Figure 1 in the future version.
>
> [2] Junxian He, Chunting Zhou, Xuezhe Ma, Taylor Berg-Kirkpatrick, and Graham Neubig. Towards a unified view of parameter-efficient transfer learning. ICLR 2022.

---

> ### Author Response · Authors · 2022-11-18
> **Thanks for your constructive feedback.**
>
> We appreciate your constructive suggestions. We provide our response to your questions as follows.
>
> **Q1: What is the training time cost of MARVEL compared to the non-adaptive methods?**
>
> We compare the training cost between MARVEL and LoRA in the following table. We use two methods to fine-tune DeBERTaV3-base on a single NVIDIA V100 GPU. We do training only and set hyperparameters, e.g., batch size and training epochs, the same as in Section 4.
>
> | Dataset | # Params | Method | GPU Mem | Time/epoch |
> | :-----: | :-----: | :-----: | :-----: | :-----: |
> | MNLI | 0.08% | LoRA | 11.094GB | 105min |
> | MNLI | 0.08% | MARVEL | 11.104GB | 116min |
> | MNLI | 0.16% | LoRA | 11.098GB | 105min |
> | MNLI | 0.16% | MARVEL | 11.110GB | 117min |
> | MNLI | 0.65% | LoRA | 11.128GB | 105min |
> | MNLI | 0.65% | MARVEL | 11.188GB | 117min |
> | SQuADv2 | 0.08% | LoRA | 13.138GB  | 60min |
> | SQuADv2 | 0.08% | MARVEL | 13.148GB | 71min |
> | SQuADv2 | 0.16% | LoRA | 13.142GB | 61min |
> | SQuADv2 | 0.16% | MARVEL | 13.164GB  | 71min |
> | SQuADv2 | 0.65% | LoRA | 13.170GB | 61min |
> | SQuADv2 | 0.65% | MARVEL | 13.226GB  | 71min |
>
> The table shows that MARVEL incurs 11% additional training time on MNLI and 16% on SQuADv2 under different budgets. The memory footprint of two methods are quite close. Such results demonstrate that MARVEL does not incur significant training overheads. The reason behind is that we only evaluate the importance score for small incremental matrices $P\Lambda Q$. Their total number of parameters is usually less than 1% of pre-trained weights. Therefore, it does not lead to significant computational cost to update the importance scores of these well-structured small matrices, compared to forward-backward pass of full model.
>
> **Q2: Impact of orthogonality constraint.**
>
> Without orthogonal regularization, the proposed SVD parameterization acts similarly as  LoRA. In that case, pruning $P\Lambda Q$ triplet-wise is similar to structured pruning on LoRA. In Section 4.4, we present an ablation study which compares MARVEL with structured pruning on LoRA (Please see Table 4).  From Table 4, we can see that MARVEL outperforms pruning LoRA on all the datasets under all budget levels.
>
> **Q3: Is the success of the gradient-weight product possibly coming from the exponential smoothing?**
>
> As Zhang, et.al., 2022 [1] mentioned, the sensitivity $I$ (Eq. (8)) estimated on mini batches incurs high uncertainty. To resolve this issue, they proposed sensitivity smoothing (exponential moving average on $I$) and uncertainty quantification. We follow their method and apply these two operations as well, which can bring benefits as shown by Table 4. In Table 4, we compare different options of importance metrics. We can see that the model performance can degenerate if defining $s(\cdot)$ without exponential moving average and uncertainty quantification $s(\cdot) = I(\cdot)$.
> For the option of $S_i = \lambda_i $ in Table 4, we do not apply exponential moving average.
>
> [1] Qingru Zhang, Simiao Zuo, Chen Liang, Alexander Bukharin, Pengcheng He, Weizhu Chen, Tuo Zhao. PLATON: Pruning Large Transformer Models with Upper Confidence Bound of Weight Importance. ICML 2022.

---

> ### Comment · Reviewer_f1zB · 2022-11-20
> **Updated Score**
>
> Thanks for the responses.
> I have updated my score from 5 -> 6.
> I am unsure if the gains (statistically significant as they are) warrant the added complexity and training overhead and hence refrain from giving an 8.

---

### Official Review · Reviewer_9ZWX · 2022-11-04

**Confidence:** 3
**Clarity, Quality, Novelty And Reproducibility:** See above
**Correctness:** 3
**Technical Novelty And Significance:** 3
**Empirical Novelty And Significance:** 3
**Recommendation:** 6

**Strength And Weaknesses:**

*Strengths*

1. The paper is well-organized and easy to follow.
2. The proposed algorithm is well-motivated. Specifically, previous work like LoRA treats all parameters equally, but actually some parameters contribute more to the final performance and thus should be updated in higher priority when the budget is limited. Therefore, the paper proposes to adaptively allocate the parameter budget among weight matrices according
to their importance score, which improves the model performance.
3. Experiments conducted on various tasks including NLU, QA, and NLG demonstrate that MARVEL can achieve better performance compared with baselines without adaptive parameter budget allocation.

*Weaknesses*

1. The experiments are only based on two backbones (DeBERTaV3-base, BART-large). It would be better to provide more results on other backbones, especially on large-scale models, e.g., T5-3B, to show the generalization of the proposed method to larger models.

**Summary Of The Paper:**

This paper proposes MARVEL, a parameter-efficient fine-tuning method. MARVEL parameterizes the incremental updates of the pre-trained weight matrices in the form of SVD for parameter-efficient fine-tuning. Besides, it allocates the parameter budget adaptively according to the importance of modules to improve the final performance. The authors conduct extensive experiments on Natural Language Understanding (NLU), Question Answering (QA) and Natural Language Generation (NLG) tasks. Results show that MARVEL outperforms existing approaches like BitFit, LoRA, etc.

**Summary Of The Review:**

In general, the paper is well-written and techniquely-reasonable. Extensive experiments demonstrate the superiority of the proposed method.

---

> ### Author Response · Authors · 2022-11-18
> **Thanks for your helpful suggestions.**
>
> **Q1: It would be better to provide more results on other backbones, especially on large-scale models.**
>
> Thanks for your helpful suggestion. We agree that the results on large-scale backbones can provide supplementary evidence of the effectiveness of our method. However, it usually requires lots of computational resources and time to fine-tune large-scale models. We can only get a few results in two weeks due to limited computational resources that are affordable for us. Specifically, we fine-tune a DeBERTaV2-xxlarge model, which contains 1.5 billion parameters, and report the validation results on RTE (Acc.) and SQuAD (EM/F1). We will post more results on other datasets and other backbones including T5-3B as they are done.
>
> | Dataset | # Params | LoRA | MARVEL |
> | :-----: | :-----: | :-----: | :-----: |
> | RTE | 0.08% | 88.81 | 91.7 |
> | RTE | 0.16% | 90.61 | 90.97 |
> | RTE | 0.32% | 90.60 | 91.34 |
> | RTE | 0.65% | 90.61 | 90.98 |
> | SQuAD | 0.65% | 89.13/94.7 | 89.84/95.14 |
>
> From the table, we can see that MARVEL outperforms LoRA under all budget levels on RTE and SQuAD. For SQuAD, we will obtain the results on the low budget levels in the following weeks.

---

### Decision · Program_Chairs · 2023-01-20

**Decision:**

Accept: poster

**Justification For Why Not Higher Score:**

The comparisons could be done with more backbone models and in more dimensions (time/memory/FLOPs...).

**Justification For Why Not Lower Score:**

The paper addresses an important problem and proposed a novel yet simple solutions.  Experiments show its effectiveness.

**Metareview: Summary, Strengths And Weaknesses:**

Summary:

This paper proposes a parameter efficient fine-tuning method which adaptively allocates the parameter budget among weight matrices.  It uses a SVD decomposition of the incremental updates of the pre-trained weight matrices, and prune unimportant triplets according to their importance scores.  Experiments are conducted on Natural Language Understanding, Question Answering and Natural Language Generation tasks which show the effectiveness of the proposed method compared to other methods with the same amount of parameters.

Strengths:

The research problem is important for applying big LMs in real applications. The proposed method is quite simple but effective and novel.  The experiments are conducted  with a comprehensive set of baselines and evaluates on a variety of tasks and show it can achieve better performance compared with baselines.   This work includes an analysis section.  The paper is well-organized and easy to follow.

Weaknesses:

The main concerns from the reviewers are on experiments, for example, more backbone models could be used and the computing resources and memory/time cost on training and inference should also be compared.  How are the hyperparameters be selected?  The writing could be improved in some parts (the SVD part for example).


**Note From Pc:**

if the above contains the word "oral" or "spotlight" please see: "oral" presentation means -> notable-top-5% and "spotlight" means -> notable-top-25%. As stated in our emails, we are disassociating presentation type from AC recommendations

**Summary Of Ac-Reviewer Meeting:**

NA